# Fast Crystal Tensor Property Prediction: A General O(3)-Equivariant Framework Based on Polar Decomposition

## Abstract

Predicting the tensor properties of crystalline materials is a fundamental task in materials science. Unlike single-value property prediction, which is inherently invariant, tensor property prediction requires maintaining $O(3)$ group tensor equivariance. This equivariance constraint often introduces tremendous computational costs, necessitating specialized designs for effective and efficient predictions. To address this limitation, we propose a general $O(3)$-equivariant framework for fast crystal tensor property prediction, called *GoeCTP*. Our framework is efficient as it does not need to impose equivalence constraints onto the network architecture. Instead, *GoeCTP* captures the tensor equivariance with a simple external rotation and reflection (R&R) module based on polar decomposition. The crafted external R&R module can rotate and reflect the crystal into an invariant standardized crystal position in space without introducing extra computational cost. We show that *GoeCTP* is general as it is a plug-and-play module that can be smoothly integrated with any existing single-value property prediction framework for predicting tensor properties. Experimental results indicate that *GoeCTP* achieves higher prediction performance and runs $13\times$ faster compared to existing state-of-the-art methods in elastic benchmarking datasets, underscoring its effectiveness and efficiency. Our code is publicly available at https://anonymous.4open.science/r/GoeCTP-FC98/.

## 1 Introduction

The tensor properties of crystalline materials can capture intricate material responses through high-order tensors, with wide-ranging applications in fields such as physics, electronics, and engineering (Yan et al., 2024b). Compared to single-value property prediction, predicting tensor properties of crystalline materials is substantially more complex. This complexity arises from the fact that tensor properties describe how crystals respond to external physical fields, such as electric fields or mechanical stress (Nye, 1985; Resta, 1994; Yan et al., 2024b). Consequently, tensor property prediction modeling necessitates preserving consistency with the crystal's spatial position, exhibiting "special" $O(3)$ equivariance (Yan et al., 2024b). Furthermore, high-order tensor property prediction is computationally intensive due to its high dimensionality and large data volume. *Therefore, providing fast and accurate predictions of tensor properties across various materials is challenging.*

Thus far, several works have been dedicated to crystal tensor property prediction. One prominent category of approaches involves *ab initio* physical simulation techniques, such as density functional theory (DFT) (Petousis et al., 2016). These classical simulation techniques can accurately calculate various material properties, including electronic structures, phonon spectra, and tensor properties. However, they necessitate extensive computational resources due to the complexity of handling a vast number of atoms and electrons in crystal systems (Yan et al., 2024b), hindering their applicability in practice. Alternatively, machine learning (ML) models have been proposed to facilitate the process of crystalline material property prediction. These methods typically leverage high-precision datasets deriving from *ab initio* simulations and utilize crystal graph construction techniques along with graph neural networks (GNNs) (Chen et al., 2019; Louis et al., 2020; Choudhary & DeCost, 2021; Xie & Grossman, 2018) or transformers (Yan et al., 2024a; Taniai et al., 2024; Yan et al., 2022; Lee et al., 2024; Wang et al., 2024a).

Existing ML methods have primarily focused on predicting single-value properties of crystals. They exhibit promising prediction performance by enforcing the periodic invariance constraint into the model design. However, these methods cannot be directly applied to tensor property prediction as they overlook the inherent anisotropy of the crystal systems, failing to achieve "special" $O(3)$ equivariance (Yan et al., 2024a; Taniai et al., 2024; Yan et al., 2022; Wang et al., 2024a). Some recent works attempt to ensure equivariance through specialized designs of network architectures (Mao et al., 2024; Lou & Ganose, 2024; Heilman et al., 2024; Wen et al., 2024; Pakornchote et al., 2023; Yan et al., 2024b; Zhong et al., 2023). These methods generally employ harmonic decomposition to achieve equivariance, where the tensor space is decomposed into the direct sum of irreducible representations of the rotation group. In this regard, numerous computationally intensive operations, such as tensor products and the merging of irreducible representations, are required. These operations incur large amounts of computation overhead, particularly when processing high-order data.

To address the above challenges, we propose a novel $O(3)$-equivariant framework for fast crystal tensor property prediction dubbed *GoeCTP*. In particular, instead of enforcing the equivariance into the model design, we craft a simple yet effective "rotation and reflection" (R&R) module based on polar decomposition. Our designed R&R module rotates and reflects the input crystal with different positions into a standardized invariant position in space. This standardized crystal position is subsequently passed into the property prediction network to obtain an invariant tensor value. Meanwhile, the orthogonal matrix obtained from the R&R module is used to achieve equivariant tensor properties prediction. Our approach is plug-and-play as it can be readily integrated with any existing single-value property prediction network for predicting tensor properties without incurring additional computational costs. Compared to the previous state-of-the-art work Yan et al. (2024b), the *GoeCTP* method achieves higher quality prediction results and runs more than $13\times$ faster in the elastic benchmarking dataset.

## 2 PRELIMINARIES AND PROBLEM STATEMENT

### 2.1 PRELIMINARIES

The structure of crystalline materials consists of a periodic arrangement of atoms in 3D space, with a repeating unit called a unit cell. An entire crystal typically can be characterized by describing the parameters of a single unit cell, such as the types and coordinates of the atoms within it, as well as the lattice parameters (Yan et al., 2022; Wang et al., 2024b; Jiao et al., 2024). General methods for describing crystals can be divided into the Cartesian coordinate system and the fractional coordinate system, as described below.

**Cartesian Coordinate System.** A crystal can be mathematically represented as $\mathbf{M} = (\mathbf{A}, \mathbf{X}, \mathbf{L})$, where $\mathbf{A} = [\boldsymbol{a}_1, \boldsymbol{a}_2, \cdots, \boldsymbol{a}_n]^T \in \mathbb{R}^{n \times d_a}$ denotes atom features for $n$ atoms within a unit cell. Each $\boldsymbol{a}_i \in \mathbb{R}^{d_a}$ is a $d_a$-dimensional feature vector characterizing an individual atom. The matrix $\mathbf{X} = [\boldsymbol{x}_1, \boldsymbol{x}_2, \cdots, \boldsymbol{x}_n]^T \in \mathbb{R}^{n \times 3}$ contains the 3D Cartesian coordinates of $n$ atoms in the unit cell. The lattice maxtrix $\mathbf{L} = [\boldsymbol{l}_1, \boldsymbol{l}_2, \boldsymbol{l}_3] \in \mathbb{R}^{3 \times 3}$ consists of the lattice vectors $\boldsymbol{l}_1$, $\boldsymbol{l}_2$, and $\boldsymbol{l}_3$, which form the basis of the 3D Euclidean space. A complete crystal is therefore represented as $(\hat{\mathbf{A}}, \hat{\mathbf{X}}) = \{(\hat{\boldsymbol{a}_i}, \hat{\boldsymbol{x}_i}) | \hat{\boldsymbol{x}_i} = \boldsymbol{x}_i + k_1 \boldsymbol{l}_1 + k_2 \boldsymbol{l}_2 + k_3 \boldsymbol{l}_3, \hat{\boldsymbol{a}_i} = \boldsymbol{a}_i, k_1, k_2, k_3 \in \mathbb{Z}, i \in \mathbb{Z}, 1 \leq i \leq n\}$. In this representation, the integers $k_i$ and $l_i$ denote all possible atomic positions in the periodic lattice.

**Fractional Coordinate System.** Instead of using the standard orthogonal basis, fractional coordinate system utilizes the lattice matrix $\mathbf{L} = [\boldsymbol{l}_1, \boldsymbol{l}_2, \boldsymbol{l}_3] \in \mathbb{R}^{3 \times 3}$ as the basis vectors for atomic positions. With this representation, the position of an atom is given by a fractional coordinate vector denoted as $\boldsymbol{f_i} = [f_1, f_2, f_3]^T \in [0, 1)^3$. The corresponding Cartesian coordinate vector can then be expressed as $\boldsymbol{x}_i = f_i \boldsymbol{l}_i$. Therefore, for a crystal $\mathbf{M}$, it can be represented as $\mathbf{M} = (\mathbf{A}, \mathbf{F}, \mathbf{L})$, where $\mathbf{F} = [\boldsymbol{f}_1, \cdots, \boldsymbol{f}_n]^T \in [0, 1)^{n \times 3}$ represents the fractional coordinates of all atoms in the unit cell.

### 2.2 PROBLEM STATEMENT

**Crystal Tensor Prediction.** The crystal tensor properties prediction is a classic regression task. Its goal is to estimate the high-order tensor property denoted as $\mathcal{Y}_{label}$ from the raw crystal data represented as $\mathbf{M} = (\mathbf{A}, \mathbf{F}, \mathbf{L})$ while minimizing the expected error between the actual property

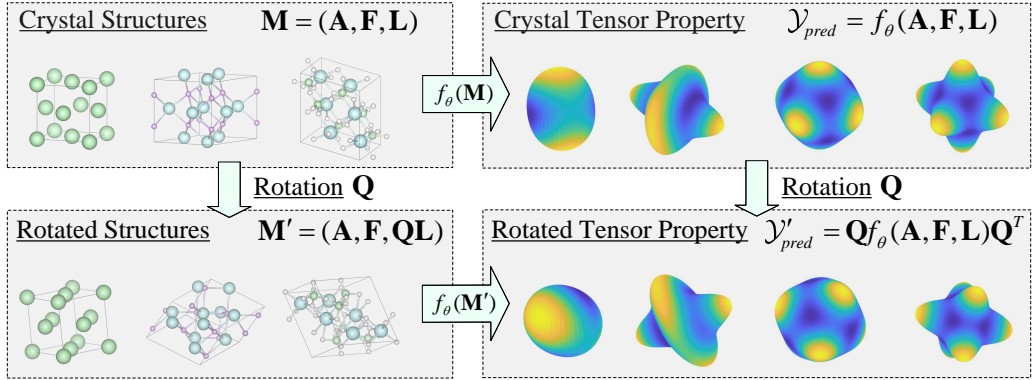

Figure 1: The illustration of $O(3)$-equivariance for crystal tensor prediction. Some of the visualizations in the figure are generated using the VESTA (Momma & Izumi, 2011), Yan et al. (2024b), and VELAS (Ran et al., 2023).

$\mathcal{Y}_{label}$ and the estimated property value $\mathcal{Y}_{pred}$. This problem can be formally formulated as follows:

$$\min_{\theta} \sum_{n=1}^{N} ||\mathcal{Y}_{pred}^{(n)} - \mathcal{Y}_{label}^{(n)}||^2, \quad \mathcal{Y}_{pred}^{(n)} = f_\theta(\mathbf{A}, \mathbf{F}, \mathbf{L}), \tag{1}$$

where $f_\theta(\cdot)$ represents a tensor prediction model with learnable parameters $\theta$, and the superscript $n$ denotes individual samples in the dataset. In what follows, we will omit superscript $n$ for simplicity. As did in the literature (Mao et al., 2024; Wen et al., 2024; Yan et al., 2024b), our objective is to estimate high-order tensor properties, including dielectric tensor (i.e., $\mathcal{Y}_{label} \triangleq \boldsymbol{\varepsilon} \in \mathbb{R}^{3\times3}$), piezoelectric tensor (i.e., $\mathcal{Y}_{label} \triangleq \mathbf{e} \in \mathbb{R}^{3\times3\times3}$), and elastic tensor (i.e., $\mathcal{Y}_{label} \triangleq C \in \mathbb{R}^{3\times3\times3\times3}$), respectively.

**O(3)-Equivariance.** The $O(3)$ group, also known as the orthogonal group in 3D, consists of all rotations and reflections in 3D space, i.e., $3 \times 3$ orthogonal matrices. In predicting the crystal tensor properties, the requirements for $O(3)$ equivariance typically differ from the $O(3)$-equivariance defined in the general molecular studies (Hoogeboom et al., 2022; Xu et al., 2024; Zheng et al., 2024; Song et al., 2024). Specifically, taking the dielectric tensor as an example where $\mathcal{Y}_{label} \triangleq \boldsymbol{\varepsilon} \in \mathbb{R}^{3\times3}$, for a tensor prediction model $f_\theta(\cdot)$ in Eq. 1, if it is $O(3)$ equivariant, it must satisfy the following equality formulated as:

$$f_\theta(\mathbf{A}, \mathbf{F}, \mathbf{QL}) = \mathbf{Q}f_\theta(\mathbf{A}, \mathbf{F}, \mathbf{L})\mathbf{Q}^T, \tag{2}$$

where $\mathbf{Q} \in \mathbb{R}^{n\times n}$ is an arbitrary orthogonal matrix (Yan et al., 2024b). For clarity, an illustration of $O(3)$-equivariance for crystal tensor prediction is shown in Fig. 1; for more equivariance details, see Appendix A.2.

**Our Objective.** Our objective is to devise a new framework that can capture the $O(3)$-equivariance for accurately predicting the high-order crystal tensor properties.

## 3 METHODOLOGY

### 3.1 THE RATIONALE OF OUR FRAMEWORK

To enforce $O(3)$ equivariance, as described in Eq. 2, existing crystal tensor prediction methods typically employ specialized network architectures based on harmonic decomposition, resulting in substantial computational overhead. For example, existing work (Yan et al., 2024b;a) gather the rotational information from neighboring nodes to the central node $i$ through the following computation formulated:

$$\boldsymbol{f}_{i,\lambda}^{l} = \frac{1}{|\mathcal{N}_i|} \sum_{j\in\mathcal{N}_i} \mathbf{TP}_\lambda(\boldsymbol{f}_j^{l'}, \mathbf{Y}_\lambda(\hat{\mathbf{e}}_{\mathbf{ji}})), \tag{3}$$

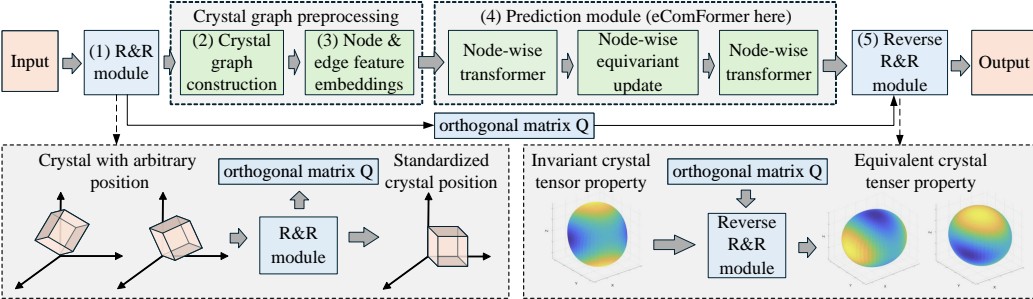

Figure 2: The Illustration of *GoeCTP*. To begin with, **(1)** the R&R module rotates and reflects the input crystal structure, which may have an arbitrary direction, to the standardized crystal position. Next, **(2)** Crystal Graph Construction module organizes the adjusted input into a crystal graph, followed by **(3)** the Node & Edge Feature Embedding module, which encodes the features of the crystal graph. Subsequently, **(4)** the Prediction module leverages these embedded features to predict the invariant tensor properties corresponding to the standardized crystal position. Finally, **(5)** the Reverse R&R module applies the orthogonal matrix $\mathbf{Q}$, obtained from the R&R module, to ensure the output of the equivariant tensor properties.

where $\boldsymbol{f}_i^l$ is node $i$'s features, $|\mathcal{N}_i|$ is the number of neighbors of node $i$, $\mathbf{TP}_\lambda$ refers to tensor product layers with rotation order $\lambda$ (Geiger & Smidt, 2022), and $\mathbf{Y}_\lambda(\hat{\mathbf{e}_{\mathbf{ji}}})$ represents the equivariant edge feature embedding, which is embedded by the corresponding spherical harmonics. When transmitting rotational information for higher-order tensors like the elastic tensor, the rotation order $\lambda$ is set to 3 (Yan et al., 2024b). In comparison, for transmitting lower-order tensor rotational information (where $\lambda$ is set to 1 (Yan et al., 2024a)), the dimension of $\mathbf{Y}_\lambda(\hat{\mathbf{e}_{\mathbf{ji}}})$ increases about double, thus, the corresponding computational cost increases. If we can transmit the rotational information and maintain the model performance for higher-order tensors with a lower value of $\lambda$, e.g., $\lambda = 1$, this would significantly reduce the computational costs.

To this end, we propose the *GoeCTP* method, which achieves $O(3)$ equivariance without requiring specialized network architecture design. For instance, it enables the transmission of rotational information for higher-order tensors using $\lambda = 1$ rather than $\lambda = 3$, thereby simplifying the model complexity while maintaining equivariance. *GoeCTP* leverages polar decomposition, a mathematical technique with significant geometric properties. Polar decomposition decomposes a matrix into two components: an orthogonal matrix, representing a rotation or reflection, and a positive semi-definite matrix (Higham, 1986). The orthogonal matrix captures an $O(3)$-group transformation, while the positive semi-definite matrix represents a stretching or scaling. When applied to crystal structures, polar decomposition effectively separates the spatial directional information of the crystal via the orthogonal matrix, allowing us to directly transfer equivariance without the need for intricate model design. Furthermore, polar decomposition is only performed during the data preprocessing stage and does not occur during network training, thereby introducing virtually no additional computational burden.

## 3.2 Our Proposed Framework: *GoeCTP*

In what follows, we will first introduce the core rotation and reflection (R&R) module of the proposed *GoeCTP*, focusing on how to obtain a standardized crystal position for a crystal with arbitrary spatial direction using polar decomposition. Then, we will describe how the input crystal data is processed and introduce the property prediction network of *GoeCTP*. Finally, we will explain how proposed *GoeCTP* achieves equivariant property predictions. An overview of the *GoeCTP* framework is illustrated in Fig. 2.

**R&R Module.** The primary function of the R&R Module is to rotate and reflect the crystal to a standardized crystal position in space. In the fractional coordinate system, the $O(3)$ group transformations applied to a crystal affect only the lattice matrix $\mathbf{L}$, while fractional coordinates remain invariant, making it convenient for the R&R module to perform rotation and reflection operations on the crystal data. Therefore, R&R module adopts the fractional coordinate system to represent the

crystal $\mathbf{M} = (\mathbf{A}, \mathbf{F}, \mathbf{L})$. This choice simplifies the implementation of the R&R module. Specifically, the application of $O(3)$-group transformations is achieved by applying polar decomposition as follows.

**Proposition 3.1** *(Polar Decomposition (Hall & Hall, 2013; Higham, 1986; Jiao et al., 2024).) An invertible matrix $\mathbf{L} \in \mathbb{R}^{3\times3}$ can be uniquely decomposed into $\mathbf{L} = \mathbf{Q}\exp(\mathbf{S})$, where $\mathbf{Q} \in \mathbb{R}^{3\times3}$ is an orthogonal matrix, $\mathbf{S} \in \mathbb{R}^{3\times3}$ is a symmetric matrix and $\exp(\mathbf{S}) = \sum_{n=0}^{\infty} \frac{\mathbf{S}^n}{n!}$ defines the exponential mapping of $\mathbf{S}$.*

Building on the above proposition, it is evident that for a crystal $\mathbf{M} = (\mathbf{A}, \mathbf{F}, \mathbf{L})$, the invertible lattice matrix $\mathbf{L}$ can be uniquely represented by a symmetric matrix $\mathbf{S}$. By defining $\mathbf{H} = \exp(\mathbf{S})$, the lattice matrix $\mathbf{L}$ can also be uniquely represented by a symmetric matrix denoted as $\mathbf{H}$, i.e. $\mathbf{L} = \mathbf{Q}\mathbf{H}$. Furthermore, $\mathbf{H}$ remains invariant with the $O(3)$ group transformation.

**Proof 3.1.1** *Given the lattice matrices $\mathbf{L}$ and $\mathbf{Q}'\mathbf{L}$, where $\mathbf{Q}'$ is an arbitrary $O(3)$ group transformation, applying polar decomposition on both $\mathbf{L}$ and $\mathbf{Q}'\mathbf{L}$ yields the following expressions:*

$$\mathbf{L} = \mathbf{Q}\mathbf{H} \tag{4}$$

$$\mathbf{Q}'\mathbf{L} = \mathbf{Q_1}\mathbf{H_1}. \tag{5}$$

*By combining Eq. 4 and 5, we can have the following equation:*

$$\mathbf{Q}'\mathbf{Q}\mathbf{H} = \mathbf{Q_1}\mathbf{H_1}. \tag{6}$$

*Because $\mathbf{Q}'\mathbf{Q}(\mathbf{Q}'\mathbf{Q})^T = \mathbf{Q}'\mathbf{Q}\mathbf{Q}^T\mathbf{Q}'^T = \mathbf{I}$, it follows that $\mathbf{Q}'\mathbf{Q}$ is an orthogonal matrix. According to the uniqueness of polar decomposition, we can have $\mathbf{Q}'\mathbf{Q} = \mathbf{Q_1}$, which implies $\mathbf{Q_1}\mathbf{H} = \mathbf{Q_1}\mathbf{H_1}$. If $\mathbf{H_1} \neq \mathbf{H}$, this would violate the uniqueness property of polar decomposition. Therefore, we conclude that $\mathbf{H_1} = \mathbf{H}$, indicating that $\mathbf{H}$ remains unaffected by the $O(3)$ group transformation.*

In essence, regardless of the $O(3)$ group transformation applied to the crystal in space, the corresponding $\mathbf{H}$ associated with $\mathbf{L}$ remains invariant. Consequently, $\mathbf{H}$ serves as a fixed standardized crystal position for the crystal $\mathbf{M} = (\mathbf{A}, \mathbf{F}, \mathbf{L})$, allowing the crystal with different position to be consistently rotated to a same position through polar decomposition. The concept of the standardized crystal position can be formally stated as the following proposition :

**Proposition 3.2** *(Standardized Crystal Position.) By performing polar decomposition on the lattice matrix $\mathbf{L} = \mathbf{Q}\mathbf{H}$, the standardized crystal position $\mathbf{H}$ is obtained. When an arbitrary $O(3)$ group transformation $\mathbf{Q}'$ is applied to a crystal structure, $\mathbf{H}$ remains unchanged.*

Therefore, the R&R Module directly applies polar decomposition to the lattice matrix $\mathbf{L}$ formulated as:

$$f_p(\mathbf{M}) = (\mathbf{A}, \mathbf{F}, \mathbf{H}), \tag{7}$$

where $f_p(\mathbf{M})$ represents the application of polar decomposition to the lattice matrix $\mathbf{L}$ of $\mathbf{M}$.

As shown in Fig. 2, the crystal $(\mathbf{A}, \mathbf{F}, \mathbf{H})$ is passed to the subsequent crystal graph construction module for further processing. The orthogonal matrix $\mathbf{Q}$ obtained during this decomposition is passed to the reverse R&R module, ensuring the equivariant transformation of the output tensor properties. Our proposed R&R module based on polar decomposition allows the input crystal data to be transformed into a standardized spatial position that is invariant under $O(3)$ space group transformations. With this particular module, the equivariance can be captured, meaning that the subsequent components of *GeoCTP* are no longer required to account for equivariance.

**Crystal Graph Construction.** To enable networks to handle such infinite crystal structures $(\mathbf{A}, \mathbf{F}, \mathbf{H})$, it is typically necessary to employ graph construction methods that represent the interactions between infinite crystal structures and atoms using finite graph data. Here, we use the crystal graph construction from Yan et al. (2024a;b) to describe the structure and relationships within crystals. Specifically, we first convert the fractional coordinate system $(\mathbf{A}, \mathbf{F}, \mathbf{H})$ into the Cartesian coordinate system $(\mathbf{A}, \mathbf{X}, \mathbf{H})$ as introduced in Sec. 2. We assume that the output crystal graph is represented as $\mathcal{G}(\mathcal{V}, \mathcal{E})$, $\mathcal{V}$ denotes the set of nodes $v_i$ in the crystal graph, where each node $v_i$ contains atomic features $\mathbf{v}_i = (\boldsymbol{a}_i, \hat{\boldsymbol{p}}_i)$. $\mathcal{E}$ represents the set of edges denoted as $e_{ij}$, which are typically constructed based on the Euclidean distance $d_{ij}$ between nodes $v_i$ and

$v_j$. When the Euclidean distance $d_{ij}$ between nodes $v_i$ and $v_j$ is less than a given radius $R$, i.e. $d_{ij} = ||\boldsymbol{p}_j + k_1\boldsymbol{l}_1 + k_2\boldsymbol{l}_2 + k_3\boldsymbol{l}_3 - \boldsymbol{p}_i||_2 \leq R$, an edge $e_{ij}$ will be built with edge feature $\mathbf{e}_{ij} = \boldsymbol{p}_j + k_1\boldsymbol{l}_1 + k_2\boldsymbol{l}_2 + k_3\boldsymbol{l}_3 - \boldsymbol{p}_i$. Here, $R$ is established based on the distance to the $k$-th nearest neighbor, and different values of $\boldsymbol{k} = [k_1, k_2, k_3] \in \mathbb{Z}^3$ represent different edges between nodes $v_i$ and $v_j$. Since there are no equivariance requirements for subsequent models, any other graph construction methods can be used to replace this part, such as Wang et al. (2024a); Yan et al. (2024a).

**Node and Edge Feature Embedding.** Building on previous work (Xie & Grossman, 2018; Yan et al., 2024b;a), node features $\boldsymbol{a}_i$ are embedded into a 92-dimensional CGCNN feature vector $\boldsymbol{f}_i$. Edge features $\mathbf{e}_{ij}$ are decomposed into their magnitude $||\mathbf{e}_{ij}||_2$ and a normalized direction vector $\hat{\mathbf{e}}_{ij}$. The magnitude is further mapped to a term similar to potential energy, $-c/||\mathbf{e}_{ij}||_2$, through the application of a radial basis function (RBF) kernel for encoding (Lin et al., 2023). Subsequently, $\mathbf{e}_{ij}$ are embedded into feature vector $\boldsymbol{f}_{ij}^e$.

**Prediction Module.** Since the R&R module is responsible for preserving equivariance, any predictive network can serve as the Prediction module, such as those proposed by Yan et al. (2024a); Taniai et al. (2024); Yan et al. (2022); Lee et al. (2024); Wang et al. (2024a), among others. For better performance, we select eComformer (Yan et al., 2024a), which has demonstrated excellent performance in single-value property prediction tasks, as our Prediction module. A detailed explanation of eComFormer can be found in Appendix A.1. Once processed through the stacked layers of eComFormer, the node features are aggregated to generate the crystal's global features as follows:

$$\boldsymbol{G}^{\text{final}} = \frac{1}{n} \sum_{1 \leq i \leq n} \boldsymbol{f}_i^{\text{final}}. \tag{8}$$

**Reverse R&R Module.** The primary function of the Reverse R&R module is to generate the equivariant tensor property predictions based on the crystal global features from the Prediction module. First, the Reverse R&R module transforms $\boldsymbol{G}^{\text{final}}$ into a tensor output, as follows:

$$\boldsymbol{\varepsilon} = f_{MLP}(\boldsymbol{G}^{\text{final}}), \tag{9}$$

$$\boldsymbol{\varepsilon}^{\text{final}} = f_{rp}(\boldsymbol{\varepsilon}, \mathbf{Q}), \tag{10}$$

where $f_{MLP}(\cdot)$ represents a multilayer perceptron (MLP) and the operation reshaping dimension. Next, $f_{rp}(\cdot)$ utilizes the orthogonal matrix $\mathbf{Q}$ obtained from the R&R Module to convert the tensor output $\boldsymbol{\varepsilon}$ into its final equivariant form denoted as $\boldsymbol{\varepsilon}^{\text{final}}$.

This conversion $f_{rp}(\cdot)$ for predicting the dielectric tensor can be expressed by:

$$\boldsymbol{\varepsilon}^{\text{final}} = \mathbf{Q}\boldsymbol{\varepsilon}\mathbf{Q}^T. \tag{11}$$

For predicting the higher-order piezoelectric and elastic tensor, the conversion process becomes more complex, seeing Appendix A.2 for more details.

## 4 RELATED WORK

**GNN-Based Methods.** CGCNN is a pioneering GNN model specifically designed for handling crystal structures (Xie & Grossman, 2018). This model proposed to represent crystal structures as multi-edge crystal graphs. It was applied to predict various single-value properties such as formation energy and band gap. Since then, several GNN methods have been developed to improve upon CGCNN through exploring various network designs or leveraging prior knowledge (Chen et al., 2019; Louis et al., 2020; Choudhary & DeCost, 2021; Das et al., 2023; Lin et al., 2023). These GNN methods are primarily designed for single-value property prediction and do not address the prediction of high-order tensor properties, such as dielectric or elastic tensors. Furthermore, they lack the ability to preserve the equivariance required for accurate high-order tensor property prediction. In contrast, recent studies attempted to ensure equivariance through specialized network architectures (Mao et al., 2024; Lou & Ganose, 2024; Heilman et al., 2024; Wen et al., 2024; Pakornchote et al., 2023; Yan et al., 2024b; Zhong et al., 2023). These approaches generally employ harmonic decomposition to achieve equivariance for tensor properties. Within these network architectures, many operations are required, such as tensor products and combining irreducible representations.

These processes significantly increase computational costs, especially when handling higher-order data.

**Transformer-Based Methods.** Transformers, with their self-attention mechanism and parallel processing capabilities, are particularly well-suited for predicting crystal material properties. Matformer (Yan et al., 2022), one of the earliest Transformer-based networks used for crystal material property prediction, encoded crystal periodic patterns by using the geometric distances between the same atoms in neighboring unit cells. This addressed the issue in earlier GNN-based methods, including CGCNN, MEGNet, GATGNN, and others, which neglected the periodic patterns of infinite crystal structures. Subsequently, more advanced Transformer-based approaches were proposed. Some methods, such as ComFormer (Yan et al., 2024a), CrystalFormer (Wang et al., 2024a), and CrystalFormer (Taniai et al., 2024), typically incorporate either enhanced graph construction techniques or physical priors, exhibiting impressive results in the prediction of single-value properties. Furthermore, DOSTransformer (Lee et al., 2024) is tailored for the density of states prediction, utilizing prompt-guided multi-modal transformer architecture to achieve super performance. Nevertheless, these models are not directly applicable for accurate high-order tensor property prediction due to their inability to capture the necessary equivariance.

# 5 EXPERIMENTS

## 5.1 EXPERIMENTAL SETUP

**Datasets.** In this work, we evaluate the performance of *GoeCTP* on three key tensor property prediction tasks: the second-order dielectric tensor, the third-order piezoelectric tensor, and the fourth-order elastic tensor, respectively. The dataset for dielectric tensor and piezoelectric tensor is derived from the data processed by Yan et al. (2024b), sourced from the JARVIS-DFT database. Similarly, the dataset for elastic tensor is obtained from the *dft_3d* data within the JARVIS-DFT database[1]. The statistical details of the datasets are presented in Table 1. Additionally, in the dielectric tensor dataset, the dielectric tensor is a $3 \times 3$ symmetric matrix. Therefore, during prediction, we predict 6 elements of the matrix (see Appendix A.3 for details) and then reconstruct the entire $3 \times 3$ symmetric matrix. In the piezoelectric tensor dataset, the piezoelectric tensor is represented using Voigt notation as a $3 \times 6$ matrix, rather than a $3 \times 3 \times 3$ third-order tensor (see Appendix A.3 for details). we only predict the $3 \times 6$ matrix. In the elastic tensor dataset, the elastic tensor is represented using Voigt notation as a $6 \times 6$ symmetric matrix, rather than a $3 \times 3 \times 3 \times 3$ fourth-order tensor (Itin & Hehl, 2013; Wen et al., 2024). Therefore, we predict the $6 \times 6$ matrix during usage (see Appendix A.3 for details).

| Dataset | Sample size | Fnorm Mean | Fnorm STD | Unit |
|---------|-------------|------------|-----------|------|
| Dielectric | 4713 | 14.7 | 18.2 | Unitless |
| Piezoelectric | 4998 | 0.43 | 3.09 | $C/m^2$ |
| Elastic | 25110 | 306.4 | 238.4 | GPa |

Table 1: Dataset statistics.

**Baseline Methods.** we selected several state-of-the-art methods in the field of crystal tensor property prediction, i.e. MEGNET (Chen et al., 2019; Morita et al., 2020), EGTNN (Zhong et al., 2023), and GMTNet (Yan et al., 2024b), as baseline methods.

**Evaluation Metrics.** We followed the evaluation metrics defined by Yan et al. (2024b) to assess the performance of the methods. The following metrics were employed: (1) **Frobenius norm (Fnorm)** is used to measure the difference between the predicted tensor and the label tensor, which is the square root of the sum of the squares of all elements in a tensor. Fnorm is widely used in various regression tasks. (2) **Error within threshold (EwT)** is determined by the ratio of the Fnorm between the predicted tensor and the ground truth tensor to the Fnorm of the ground truth tensor. This ratio can be expressed as $||y_{pred} - y_{label}||_F / ||y_{label}||_F$, where $|| \cdot ||_F$ is Fnorm, and $y_{label}$ and $y_{pred}$ represent the ground truth and predicted values, respectively. For instance, EwT 25% indicates that the proportion of predicted samples with this ratio is below 25%. Higher values of EwT signify

---

[1]https://pages.nist.gov/jarvis/databases/

better prediction quality. In our experiments, we utilized several thresholds for EwT: EwT 25%, EwT 10%, and EwT 5%.

**Experimental Settings.** The experiments were performed using an NVIDIA GeForce RTX 3090 GPU. For benchmarking, we directly utilized the codebases for MEGNET, ETGNN, and GMTNet as provided by Yan et al. (2024b). For each property, the dataset is split into training, validation, and test sets in an 8:1:1 ratio. During model training, Huber loss, AdamW (Loshchilov & Hutter, 2018), a weight decay of $10^{-5}$, and polynomial learning rate decay were employed.

Since we observed some randomness in the results during the training process of the repeated dielectric dataset experiments, we conducted 5 repeated experiments for each method and used the average of the 5 experiment metrics as the final experimental results. The results of the 5 repeated experiments can be found in the Appendix A.4.

## 5.2 Experimental Results

**Predicting Dielectric Tensors.** The performance of various models in predicting the dielectric tensor is summarized in Table 2. While *GoeCTP* and GMTNet show close results in terms of Fnorm and EwT 25%, *GoeCTP* demonstrates higher values for EwT 10% and EwT 5%, with an improvement of 5% in EwT 5% compared to GMTNet. This indicates that *GoeCTP* delivers higher-quality predictions compared to GMTNet. Furthermore, as *GoeCTP* is designed to be a flexible framework, we also evaluated its combination with other models, such as iComFormer (Yan et al., 2024a) and CrystalFormer (Taniai et al., 2024). Detailed results of these combinations are provided in Appendix A.4.

|  | MEGNET | ETGNN | GMTNet | GoeCTP (Ours) |
|---|---|---|---|---|
| Fnorm ↓ | 3.71 | 3.40 | 3.28 | **3.23** |
| EwT 25% ↑ | 75.8% | 82.6% | **83.3%** | 83.2% |
| EwT 10% ↑ | 38.9% | 49.1% | 56.0% | **56.8%** |
| EwT 5% ↑ | 18.0% | 25.3% | 30.5% | **35.5%** |

Table 2: Comparison of performance metrics between MEGNET, ETGNN, GMTNet, and GoeCTP on the dielectric dataset.

**Predicting Piezoelectric tensors.** The experimental results for the elastic tensor dataset are shown in Table 3. Similar to the experimental results on the dielectric tensors dataset, although the EwT 25% value of *GoeCTP* is not as good as GMTNet, it achieves optimal results on all other evaluation metrics.

|  | MEGNET | ETGNN | GMTNet | GoeCTP (Ours) |
|---|---|---|---|---|
| Fnorm ↓ | 0.465 | 0.535 | 0.462 | **0.448** |
| EwT 25% ↑ | 43.9% | 37.5% | **45.7%** | 44.9% |
| EwT 10% ↑ | 37.9% | 22.8% | 39.3% | **43.1%** |
| EwT 5% ↑ | 27.1% | 13.8% | 35.7% | **40.1%** |

Table 3: Comparison of performance metrics between MEGNET, ETGNN, GMTNet, and GoeCTP on the piezoelectric dataset.

**Predicting Elastic Tensors.** Experimental results on the elastic tensor dataset are presented in Table 4. *GoeCTP* demonstrated outstanding performance in predicting higher-order, complex tensors, outperforming all baseline methods across every evaluation metric. Notably, it achieved the lowest Fnorm of 107.11 and improved all EwT metrics by an average of 6% compared to GMTNet, emphasizing its general applicability in predicting diverse tensor properties in materials science. Further details on the results of *GoeCTP*'s combination with other models can be found in Appendix A.4.

**Verifying the $O(3)$ Equivariance.** To evaluate the effectiveness of *GoeCTP*, we conducted experiments to verify the $O(3)$ equivariance of tensor properties. Specifically, after training *GoeCTP*, we extracted its Prediction module (i.e., eComFormer) for comparative testing on two different test

|  | MEGNET | ETGNN | GMTNet | GoeCTP (Ours) |
|---|---|---|---|---|
| Fnorm ↓ | 143.86 | 123.64 | 117.62 | **107.11** |
| EwT 25% ↑ | 23.6% | 32.0% | 36.0% | **42.5%** |
| EwT 10% ↑ | 3.0% | 3.8% | 7.6% | **15.3%** |
| EwT 5% ↑ | 0.5% | 0.5% | 2.0% | **7.2%** |

Table 4: Comparison of performance metrics between MEGNET, ETGNN, GMTNet, and GoeCTP on the elastic dataset.

sets (original test set and augmented test set). All crystals in the original test set were adjusted to the standardized crystal position, while the augmented test set was generated by applying arbitrary $O(3)$ group transformations to all crystals in the original test set. The method for generating the corresponding orthogonal matrices is from Heiberger (1978). We then evaluated both the Prediction module (eComFormer) and *GoeCTP* on these two datasets and compared the performance metrics.

The results on the dielectric tensor dataset are shown in Table 5. *GoeCTP* performed equally well on the augmented test set as on the original test set, indicating that it maintains strong $O(3)$ equivariance for tensor properties. In contrast, the performance of the eComFormer method significantly declined on the augmented test set, with Fnorm decreasing by nearly 45%, demonstrating that it does not meet the $O(3)$ equivariance requirements for tensor properties. The results on the piezoelectric tensor dataset are presented in Table 6. Similar to the dielectric tensor dataset, *GoeCTP* maintained consistent performance between the augmented and original test sets. In contrast, the eComFormer method exhibited a decline in performance on the augmented test set, indicating its inability to fully satisfy the $O(3)$ equivariance requirements for tensor properties. The results on the elastic tensor dataset are shown in Table 7. Similar to the dielectric tensor dataset, the performance of the eComFormer method significantly declined on the augmented test set, with Fnorm dropping by nearly 29%. In contrast, *GoeCTP*'s performance remained unchanged, further demonstrating the effectiveness of the *GoeCTP* method. Additionally, runtime comparisons between *GoeCTP* and eComFormer on the test set showed that *GoeCTP* introduced no significant increase in runtime. This indicates that when integrated into single-value property prediction networks, *GoeCTP* incurs almost no additional computational cost, enhancing its practicality without compromising efficiency.

|  | eCom. (ori. data) | eCom. (aug. data) | **GoeCTP (Ours)** (ori. data) | **GoeCTP (Ours)** (aug. data) |
|---|---|---|---|---|
| Fnorm ↓ | 3.23 | 4.71 | 3.23 | 3.23 |
| EwT 25% ↑ | 83.2% | 69.7% | 83.2% | 83.2% |
| EwT 10% ↑ | 56.8% | 42.7% | 56.8% | 56.8% |
| EwT 5% ↑ | 35.5% | 22.5% | 35.5% | 35.5% |
| Total Time (s) ↓ | 26.03 | 26.01 | 26.23 | 26.18 |

Table 5: Ablation study for verifying the $O(3)$ equivariance with dielectric dataset.

|  | eCom. (ori. data) | eCom. (aug. data) | **GoeCTP (Ours)** (ori. data) | **GoeCTP (Ours)** (aug. data) |
|---|---|---|---|---|
| Fnorm ↓ | 0.448 | 0.496 | 0.448 | 0.448 |
| EwT 25% ↑ | 44.9% | 44.3% | 44.9% | 44.9% |
| EwT 10% ↑ | 43.1% | 42.1% | 43.1% | 43.1% |
| EwT 5% ↑ | 40.1% | 38.3% | 40.1% | 40.1% |
| Total Time (s) ↓ | 12.83 | 12.23 | 13.01 | 12.71 |

Table 6: Ablation study for verifying the $O(3)$ equivariance with piezoelectric dataset.

**Efficiency.** The results presented in Table 8, Table 9, and Table 10 illustrate the running times for *GoeCTP* and baseline methods. On the dielectric dataset, *GoeCTP* completed the entire training process with only 38.2% of the time spent compared to GMTNet. On the piezoelectric dataset, *GoeCTP* completed the entire training process with only 16.3% of the time spent compared to GMTNet. On the elastic dataset, *GoeCTP* required less than 7.0% of the time spent by GMTNet to complete the entire training process. To achieve $O(3)$ equivariance for tensor properties, GMTNet's network architecture relies on irreducible representations and tensor operations, which considerably reduce computational efficiency, especially for higher-order tensor property (elastic tensor) prediction tasks

| | eCom. (ori. data) | eCom. (aug. data) | GoeCTP (Ours) (ori. data) | GoeCTP (Ours) (aug. data) |
|---|---|---|---|---|
| Fnorm ↓ | 107.11 | 138.45 | 107.11 | 107.11 |
| EwT 25% ↑ | 42.5% | 25.9% | 42.5% | 42.5% |
| EwT 10% ↑ | 15.3% | 2.2% | 15.3% | 15.3% |
| EwT 5% ↑ | 7.2% | 0.2% | 7.2% | 7.2% |
| Total Time (s) ↓ | 83.26 | 83.02 | 90.10 | 89.60 |

Table 7: Ablation study for verifying the $O(3)$ equivariance with elastic dataset.

where the time cost of GMTNet increases sharply. In contrast, *GoeCTP* requires no special architectural design for tensor property prediction tasks; it predicts tensor properties of varying orders using only a MLP at the network's output, ensuring both efficiency and scalability across different tensor orders.

| | MEGNET | ETGNN | GMTNet | GoeCTP (Ours) |
|---|---|---|---|---|
| Total Time (s) ↓ | 663 | 1325 | 1611 | **616** |
| Time/batch (s) ↓ | 0.052 | 0.104 | 0.126 | **0.048** |

Table 8: Efficiency comparison on the dielectric dataset.

| | MEGNET | ETGNN | GMTNet | GoeCTP (Ours) |
|---|---|---|---|---|
| Total Time (s) ↓ | **843** | 1220 | 5771 | 938 |
| Time/batch (s) ↓ | **0.065** | 0.095 | 0.45 | 0.073 |

Table 9: Efficiency comparison on the piezoelectric dataset.

| | MEGNET | ETGNN | GMTNet | GoeCTP (Ours) |
|---|---|---|---|---|
| Total Time (s) ↓ | 2899 | 4448 | > 36000 | **2422** |
| Time/batch (s) ↓ | 0.226 | 0.348 | > 2.813 | **0.189** |

Table 10: Efficiency comparison on the elastic dataset.

## 6 CONCLUSION, LIMITATIONS, AND FUTURE WORKS

In this work, we propose a novel $O(3)$-equivariant framework *GoeCTP* for fast crystal tensor prediction, which is a plug-and-play framework that can be integrated with any existing single-value property prediction network, enabling them to predict tensor properties with almost no additional computational cost. Based on predicting invariant tensor properties from standardized crystal positions and using an external module to ensure tensor equivariance, this approach has achieved state-of-the-art performance and the highest efficiency in different widely used crystal tensor property benchmark tests. The limitations of our current *GoeCTP* include (1) The performance of *GoeCTP* is inherently dependent on the performance of the Prediction module. (2) Currently, *GoeCTP* is specifically tailored for the prediction of tensor properties in crystalline materials and has not yet been adapted for other material types. In future work, we intend to explore the following directions: for (1), we aim to improve performance by integrating prior knowledge related to the independent components of tensor properties across different crystal systems in Appendix A.3 (Further discussion on the utilization of space group constraints on tensor properties can be found in Appendix A.6); for (2), we plan to extend the framework to other domains, such as 3D point cloud research, broadening its applicability. Seeing Appendix A.5 for more extensibility and limitations details.

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

# A APPENDIX

## A.1 DETAILS OF ECOMFORMER

The eComFormer has demonstrated strong performance across a range of single-value property prediction tasks (Yan et al., 2024a). It is built on an SO(3)-equivariant crystal graph representation, where the interatomic distance vectors are employed to represent the edge features of the graph. The model converts this crystal graph into embeddings and utilizes a transformer architecture, incorporating both a node-wise transformer layer and a node-wise equivariant updating layer, to extract rich geometric information during the message-passing process.

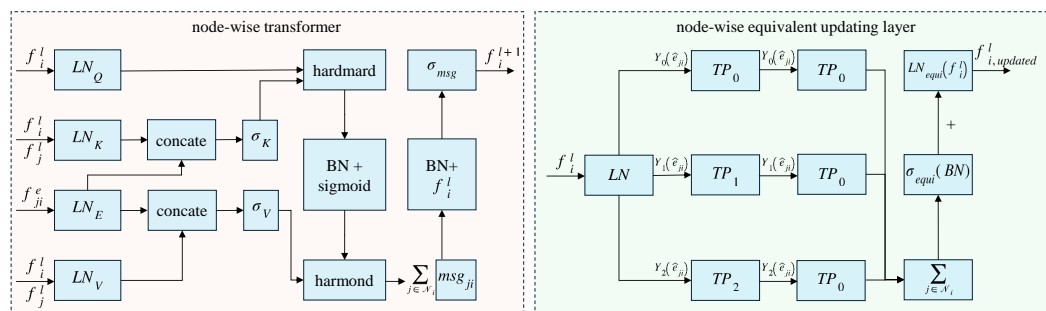

Figure 3: The detailed architectures of the node-wise transformer layer and node-wise equivariant updating layer, adapted from Yan et al. (2024a).

Specifically, the node-wise transformer layer is responsible for updating the node-invariant features $\boldsymbol{f}_i$. This process utilizes the node features $\boldsymbol{f}_i$, neighboring node features $\boldsymbol{f}_j$, and edge features $\boldsymbol{f}_{ij}$ to facilitate message passing from neighboring node $j$ to the central node $i$, followed by aggregation of all neighboring messages to update $\boldsymbol{f}_i$. The update mechanism is structured similarly to a transformer. Fristly, the message from node $j$ to node $i$ is transformed into corresponding query $\boldsymbol{q}_{ij} = \mathrm{LN}_Q(\boldsymbol{f}_i)$, key $\boldsymbol{k}_{ij} = (\mathrm{LN}_K(\boldsymbol{f}_i)|\mathrm{LN}_K(\boldsymbol{f}_j))$, and value feature $\boldsymbol{v}_{ij} =$

$(\mathrm{LN}_V(\boldsymbol{f}_i)|\mathrm{LN}_V(\boldsymbol{f}_j)|\mathrm{LN}_E(\boldsymbol{f}_{ij}^e))$, where $\mathrm{LN}_Q(\cdot)$, $\mathrm{LN}_K(\cdot)$, $\mathrm{LN}_V(\cdot)$, $\mathrm{LN}_E(\cdot)$ denote the linear transformations, and $|$ denote the concatenation. Then, the self-attention output is then computed as:

$$\boldsymbol{\alpha}_{ij} = \frac{\boldsymbol{q}_{ij} \circ \boldsymbol{\xi}_K(\boldsymbol{k}_{ij})}{\sqrt{d_{\boldsymbol{q}_{ij}}}}, \boldsymbol{msg}_{ij} = \mathrm{sigmoid}(\mathrm{BN}(\boldsymbol{\alpha}_{ij})) \circ \boldsymbol{\xi}_V(\boldsymbol{v}_{ij}), \tag{12}$$

where $\boldsymbol{\xi}_K$, $\boldsymbol{\xi}_V$ represent nonlinear transformations applied to key and value features, respectively, and the operators $\circ$ denote the Hadamard product, $\boldsymbol{\xi}_{msg}(\cdot)$ refers to the batch normalization layer, and $\sqrt{d_{\boldsymbol{q}_{ij}}}$ indicates the dimensionality of $\boldsymbol{q}_{ij}$. Then, node feature $\boldsymbol{f}_i$ is updated as follows,

$$\boldsymbol{msg}_i = \sum_{j \in \mathcal{N}_i} \boldsymbol{msg}_{ij}, \boldsymbol{f}_i^{\mathrm{new}} = \boldsymbol{\xi}_{msg}(\boldsymbol{f}_i + \mathrm{BN}(\boldsymbol{msg}_i)), \tag{13}$$

where $\boldsymbol{\xi}_{msg}(\cdot)$ denoting the softplus activation function.

The node-wise equivariant updating layer is designed to effectively capture geometric features by incorporating node feature $\mathbf{a}_i$ and edge feature $||\mathbf{e}_{ji}||_2$ as inputs and stacking two tensor product (TP) layers (Geiger & Smidt, 2022). It uses node feature $\boldsymbol{f}_i^l$ and equivalent vector feature $\mathbf{e}_{ji}$ embedded by corresponding spherical harmonics $\mathbf{Y}_0(\hat{\mathbf{e}}_{ji}) = c_0$, $\mathbf{Y}_1(\hat{\mathbf{e}}_{ji}) = c_1 * \frac{\mathbf{e}_{ji}}{||\mathbf{e}_{ji}||_2} \in \mathbb{R}^3$ and $\mathbf{Y}_2(\hat{\mathbf{e}}_{ji}) \in \mathbb{R}^5$ to represent the input features. Gathering rotational information from neighboring nodes to the central node $i$, the first TP layer is shown as

$$\boldsymbol{f}_{i,0}^l = \boldsymbol{f}_i^{l'} + \frac{1}{|\mathcal{N}_i|} \sum_{j \in \mathcal{N}_i} \mathbf{TP}_0(\boldsymbol{f}_j^{l'}, \mathbf{Y}_0(\hat{\mathbf{e}_{ji}})), \boldsymbol{f}_{i,\lambda}^l = \frac{1}{|\mathcal{N}_i|} \sum_{j \in \mathcal{N}_i} \mathbf{TP}_\lambda(\boldsymbol{f}_j^{l'}, \mathbf{Y}_\lambda(\hat{\mathbf{e}_{ji}})), \lambda \in \{1, 2\}, \tag{14}$$

where $\boldsymbol{f}_i^{l'}$ is linearly transformed from $\boldsymbol{f}_i^l$, $|\mathcal{N}_i|$ and $\mathbf{TP}_\lambda$ represent the number of neighbors of node $i$ and TP layers with rotation order $\lambda$ respectively. Then, to represent the invariant node features, the second TP layer is written as

$$\boldsymbol{f}_i^{l*} = \frac{1}{|\mathcal{N}_i|}(\sum_{j \in \mathcal{N}_i} \mathbf{TP}_0(\boldsymbol{f}_{j,0}^l, \mathbf{Y}_0(\hat{\mathbf{e}_{ji}})) + \sum_{j \in \mathcal{N}_i} \mathbf{TP}_0(\boldsymbol{f}_{j,1}^l, \mathbf{Y}_1(\hat{\mathbf{e}_{ji}})) + \sum_{j \in \mathcal{N}_i} \mathbf{TP}_0(\boldsymbol{f}_{j,2}^l, \mathbf{Y}_2(\hat{\mathbf{e}_{ji}}))), \tag{15}$$

stacking the two tensor product layers together using both linear and nonlinear transformations, the output $\boldsymbol{f}_{i,updated}^l$ is combined as

$$\boldsymbol{f}_{i,updated}^l = \boldsymbol{\sigma}_{\mathrm{equi}}(\mathrm{BN}(\boldsymbol{f}_i^{l*})) + \mathrm{LN}_{\mathrm{equi}}(\boldsymbol{f}_i^l), \tag{16}$$

with $\boldsymbol{\sigma}_{\mathrm{equi}}$ denoting a nonlinear transformation made up of two softplus layers with a linear layer positioned between them. The detailed architectures of the node-wise transformer layer and node-wise equivariant updating layer are shown in Fig. 3.

## A.2   $O(3)$ EQUIVARIANCE FOR CRYSTAL TENSOR PROPERTIES

The $O(3)$ equivariance required for crystal tensor property prediction tasks differs from the $O(3)$ equivariance typically encountered in general molecular studies (Hoogeboom et al., 2022; Xu et al., 2024; Song et al., 2024). In molecular studies, for a function $f : (\mathbf{A}, \mathbf{F}, \mathbf{L}) \to \mathbf{y} \in \mathbb{R}^n$, if it is $O(3)$ equivariant, then it satisfies $f(\mathbf{A}, \mathbf{F}, \mathbf{QL}) = \mathbf{Q}f(\mathbf{A}, \mathbf{F}, \mathbf{L})$, where $\mathbf{Q} \in \mathbb{R}^{n \times n}$ is an arbitrary orthogonal matrix. Here, the function $f$ can be seen as the model.

**Dielectric tensor.** However, in the prediction of crystal tensor properties, such as the dielectric tensor $\boldsymbol{\varepsilon} \in \mathbb{R}^{3 \times 3}$, the requirements for $O(3)$ equivariance differ. For a function $f : (\mathbf{A}, \mathbf{F}, \mathbf{L}) \to \boldsymbol{\varepsilon}$, if it is $O(3)$ equivariant, it must satisfy $f(\mathbf{A}, \mathbf{F}, \mathbf{QL}) = \mathbf{Q}f(\mathbf{A}, \mathbf{F}, \mathbf{L})\mathbf{Q}^T$. The specific reason for this difference can be referenced in the previous work (Yan et al., 2024b), and is rooted in the fact that the dielectric tensor characterizes a material's polarization response to an external electric field, describing the relationship between the electric displacement $\mathbf{D} \in \mathbb{R}^3$ and the applied electric field $\mathbf{E} \in \mathbb{R}^3$ by $\mathbf{D} = \boldsymbol{\varepsilon}\mathbf{E}$. When an $O(3)$ group transformation $\mathbf{Q}$ is applied to the crystal structure, we have $\mathbf{D}' = \boldsymbol{\varepsilon}'\mathbf{E}'$, where $\mathbf{D}' = \mathbf{QD}$ and $\mathbf{E}' = \mathbf{QE}$. This results in the transformation of the dielectric tensor under $O(3)$ group transformation as $\boldsymbol{\varepsilon}' = \mathbf{Q}\boldsymbol{\varepsilon}\mathbf{Q}^T$. This transformation principle extends similarly to other crystal tensors, as follows.

**Piezoelectric tensor.** The piezoelectric tensor $\mathbf{e} \in \mathbb{R}^{3 \times 3 \times 3}$, describes the relationship between the applied strain $\boldsymbol{\epsilon} \in \mathbb{R}^{3 \times 3}$ to the electric displacement field $\mathbf{D} \in \mathbb{R}^3$ within the material. Mathematically, this relationship is expressed as $\mathbf{D}_i = \sum_{jk} \mathbf{e}_{ijk} \boldsymbol{\epsilon}_{jk}$, with $i, j, k \in \{1, 2, 3\}$. When an $O(3)$ group transformation $\mathbf{Q}$ is applied to the crystal, the strain tensor and electric displacement field is transformed to $\boldsymbol{\epsilon}'_{jk} = \sum_{mn} \mathbf{Q}_{jm} \mathbf{Q}_{kn} \boldsymbol{\epsilon}_{mn}$ and $\mathbf{D}'_i = \sum_{\ell} \mathbf{Q}_{i\ell} \mathbf{D}_{\ell}$. The relation is then reformulated as $\mathbf{D}'_i = \sum_{jk} \mathbf{e}'_{ijk} \boldsymbol{\epsilon}'_{jk}$.

Since $\mathbf{Q}$ is orthogonal matrix ($\mathbf{Q}^{-1} = \mathbf{Q}^T$), we have $\boldsymbol{\epsilon}_{jk} = \sum_{mn} \mathbf{Q}_{mj} \mathbf{Q}_{nk} \boldsymbol{\epsilon}'_{mn}$. Consequently, $\mathbf{D}'_i$ can be represented as

$$
\begin{aligned}
\mathbf{D}'_i &= \sum_{\ell} \mathbf{Q}_{i\ell} \mathbf{D}_{\ell} \\
&= \sum_{\ell} \mathbf{Q}_{i\ell} \sum_{jk} \mathbf{e}_{\ell jk} \boldsymbol{\epsilon}_{jk} \\
&= \sum_{\ell} \mathbf{Q}_{i\ell} \sum_{jk} \mathbf{e}_{\ell jk} (\sum_{mn} \mathbf{Q}_{mj} \mathbf{Q}_{nk} \boldsymbol{\epsilon}'_{mn}) \\
&= \sum_{\ell} \mathbf{Q}_{i\ell} \sum_{mn} \mathbf{e}_{\ell mn} (\sum_{jk} \mathbf{Q}_{jm} \mathbf{Q}_{kn} \boldsymbol{\epsilon}'_{jk}) \quad (exchange\, sign, m \leftrightarrow j, n \leftrightarrow k) \\
&= \sum_{jk} \sum_{lmn} \mathbf{Q}_{il} \mathbf{Q}_{jm} \mathbf{Q}_{kn} \mathbf{e}_{lmn} \boldsymbol{\epsilon}'_{jk}
\end{aligned}
\tag{17}
$$

Therefore, under the $O(3)$ group transformation $\mathbf{Q}$, the transformed piezoelectric tensor $\mathbf{e}'_{ijk}$ is given by:

$$
\mathbf{e}'_{ijk} = \sum_{lmn} \mathbf{Q}_{il} \mathbf{Q}_{jm} \mathbf{Q}_{kn} \mathbf{e}_{lmn}.
\tag{18}
$$

When employing GoeCTP to predict the piezoelectric tensor, Equation 11 becomes $\mathbf{e}^{\text{final}}_{ijk} = \sum_{lmn} \mathbf{Q}_{il} \mathbf{Q}_{jm} \mathbf{Q}_{kn} \mathbf{e}_{lmn}$.

**Elastic tensor.** The elastic tensor $C \in \mathbb{R}^{3 \times 3 \times 3 \times 3}$ relates the applied strain $\boldsymbol{\epsilon} \in \mathbb{R}^{3 \times 3}$ to the stress tensor $\sigma \in \mathbb{R}^{3 \times 3}$ within the material, expressed as: $\boldsymbol{\sigma}_{ij} = \sum_{k\ell} C_{ijk\ell} \boldsymbol{\epsilon}_{k\ell}$, with $i, j, k, \ell \in \{1, 2, 3, 4\}$. When an $O(3)$ group transformation $\mathbf{Q}$ is applied to the crystal, the strain tensor and stress tensor is transformed to $\boldsymbol{\epsilon}'_{jk} = \sum_{mn} \mathbf{Q}_{jm} \mathbf{Q}_{kn} \boldsymbol{\epsilon}_{mn}$ and $\boldsymbol{\sigma}'_{jk} = \sum_{mn} \mathbf{Q}_{jm} \mathbf{Q}_{kn} \boldsymbol{\sigma}_{mn}$. The relation becomes $\boldsymbol{\sigma}'_{ij} = \sum_{k\ell} C'_{ijk\ell} \boldsymbol{\epsilon}'_{k\ell}$.

Because $\mathbf{Q}$ is orthogonal matrix ($\mathbf{Q}^{-1} = \mathbf{Q}^T$), we have $\boldsymbol{\epsilon}_{jk} = \sum_{mn} \mathbf{Q}_{mj} \mathbf{Q}_{nk} \boldsymbol{\epsilon}'_{mn}$, then $\boldsymbol{\sigma}'_{ij}$ can be represented as

$$
\begin{aligned}
\boldsymbol{\sigma}'_{ij} &= \sum_{mn} \mathbf{Q}_{im} \mathbf{Q}_{jn} \boldsymbol{\sigma}_{mn} \\
&= \sum_{mn} \mathbf{Q}_{im} \mathbf{Q}_{jn} \sum_{pq} C_{mnpq} \boldsymbol{\epsilon}_{pq} \\
&= \sum_{mn} \mathbf{Q}_{im} \mathbf{Q}_{jn} \sum_{pq} C_{mnpq} \sum_{k\ell} \mathbf{Q}_{kp} \mathbf{Q}_{\ell q} \boldsymbol{\epsilon}'_{k\ell} \\
&= \sum_{k\ell} \sum_{mnpq} \mathbf{Q}_{im} \mathbf{Q}_{jn} \mathbf{Q}_{kp} \mathbf{Q}_{lq} C_{mnpq} \boldsymbol{\epsilon}'_{k\ell}
\end{aligned}
\tag{19}
$$

Therefore, under the $O(3)$ group transformation $\mathbf{Q}$, $C'_{ijkl}$ is represented as:

$$
C'_{ijkl} = \sum_{mnpq} \mathbf{Q}_{im} \mathbf{Q}_{jn} \mathbf{Q}_{kp} \mathbf{Q}_{lq} C_{mnpq}.
\tag{20}
$$

When GoeCTP is used to predict elastic tensor, Equation 11 becomes $C^{\text{final}}_{ijkl} = \sum_{mnpq} \mathbf{Q}_{im} \mathbf{Q}_{jn} \mathbf{Q}_{kp} \mathbf{Q}_{lq} C_{mnpq}$.

### A.3 TENSOR PROPERTIES SYMMETRY

Tensor properties, such as dielectric tensors and elastic tensors, describe the material's response to external physical fields (such as electric fields or mechanical stress). In materials with symmetry, this response must adhere to the symmetry requirements of the material. As demonstrated by Yan et al. (2024b), when the space group transformation $\mathbf{R}$ is applied to the corresponding crystal structure $\mathbf{M} = (\mathbf{A}, \mathbf{F}, \mathbf{L})$, the crystal remains unchanged, i.e., $(\mathbf{A}, \mathbf{F}, \mathbf{L}) = (\mathbf{A}, \mathbf{F}, \mathbf{RL})$. Therefore, the corresponding tensor properties also remain unchanged, i.e., $\varepsilon = \mathbf{R}\varepsilon\mathbf{R}^T$. Thus, crystal symmetry imposes strict constraints on the components of the tensor, leading to the simplification or elimination of many components, reducing the number of independent components.

| Crystal system | Number of independent elements | Dielectric tensor |
|---|---|---|
| Cubic | 1 | $\varepsilon = \begin{pmatrix} \varepsilon_{11} & 0 & 0 \\ 0 & \varepsilon_{11} & 0 \\ 0 & 0 & \varepsilon_{11} \end{pmatrix}$ |
| Tetragonal & Hexagonal & Trigonal | 2 | $\varepsilon = \begin{pmatrix} \varepsilon_{11} & 0 & 0 \\ 0 & \varepsilon_{11} & 0 \\ 0 & 0 & \varepsilon_{33} \end{pmatrix}$ |
| Orthorhombic | 3 | $\varepsilon = \begin{pmatrix} \varepsilon_{11} & 0 & 0 \\ 0 & \varepsilon_{22} & 0 \\ 0 & 0 & \varepsilon_{33} \end{pmatrix}$ |
| Monoclinic | 4 | $\varepsilon = \begin{pmatrix} \varepsilon_{11} & 0 & \varepsilon_{13} \\ 0 & \varepsilon_{22} & 0 \\ \varepsilon_{13} & 0 & \varepsilon_{33} \end{pmatrix}$ |
| Triclinic | 6 | $\varepsilon = \begin{pmatrix} \varepsilon_{11} & \varepsilon_{12} & \varepsilon_{13} \\ \varepsilon_{12} & \varepsilon_{22} & \varepsilon_{23} \\ \varepsilon_{13} & \varepsilon_{23} & \varepsilon_{33} \end{pmatrix}$ |

Table 11: Number of independent components in the dielectric tensor for different crystal systems.

**Independent components in the dielectric tensor.** The $3 \times 3$ dielectric tensor has a minimum of 1 and a maximum of 6 independent elements for various types of systems due to the crystal symmetry. The number of independent components in the dielectric tensor for different crystal systems is shown in Tabel 11 (Mao et al., 2024).

**Voigt notation for elastic tensor.** Similar to the dielectric tensor, the elastic tensor also has independent elements for various types of systems due to the crystal symmetry. The elastic tensor has a minimum of 3 and a maximum of 21 independent elements for various types of systems. Voigt notation is a compact way to represent these independent components of tensor properties (Itin & Hehl, 2013). According to the rules $11 \rightarrow 1\,;\,22 \rightarrow 2\,;\,33 \rightarrow 3\,;\,23,\ 32 \rightarrow 4\,;\,31,\ 13 \rightarrow 5\,;\,12,\ 21 \rightarrow 6$, the elastic tensor in Voigt notation is a $6 \times 6$ symmetric matrix (Wen et al., 2024; Ran et al., 2023):

$$C = \begin{pmatrix} C_{1111} & C_{1122} & C_{1133} & C_{1123} & C_{1131} & C_{1112} \\ C_{1122} & C_{2222} & C_{2233} & C_{2223} & C_{2231} & C_{2212} \\ C_{1133} & C_{2233} & C_{3333} & C_{3323} & C_{3331} & C_{3312} \\ C_{1123} & C_{2223} & C_{3323} & C_{2323} & C_{2331} & C_{2312} \\ C_{1131} & C_{2231} & C_{3331} & C_{2331} & C_{3131} & C_{3112} \\ C_{1112} & C_{2212} & C_{3312} & C_{2312} & C_{3112} & C_{1212} \end{pmatrix} \rightarrow \begin{pmatrix} C_{11} & C_{12} & C_{13} & C_{14} & C_{15} & C_{16} \\ C_{12} & C_{22} & C_{23} & C_{24} & C_{25} & C_{26} \\ C_{13} & C_{23} & C_{33} & C_{34} & C_{35} & C_{36} \\ C_{14} & C_{24} & C_{34} & C_{44} & C_{45} & C_{46} \\ C_{15} & C_{25} & C_{35} & C_{45} & C_{55} & C_{56} \\ C_{16} & C_{26} & C_{36} & C_{46} & C_{56} & C_{66} \end{pmatrix} \quad (21)$$

The number of independent components in the elastic tensor for different crystal systems is shown in Tabel 12 (partial data shown; for more details, refer to Wen et al. (2024); Ran et al. (2023)).

**Voigt notation for piezoelectric tensor.** The number of independent components in the piezoelectric tensor for different crystal systems is shown in Tabel 13 (partial data shown; for more details, refer to De Jong et al. (2015); Gorfman & Zhang (2024)).

### A.4 EXPERIMENTAL DETAILS AND ADDITIONAL RESULTS

**Hyperparameter settings of GoeCTP.** When constructing the crystal graph, we used the 16th nearest atom to determine the cutoff radius. For edge embeddings, we used an RBF kernel with $c = 0.75$ and values ranging from $-4$ to $0$, which maps $-c/||\mathbf{e}_{ij}||_2$ to a 512-dimensional vector. In the

| Crystal system | Number of independent elements | Elastic tensor |
|---|---|---|
| Cubic | 3 | $C = \begin{pmatrix} C_{11} & C_{12} & C_{12} & 0 & 0 & 0 \\ C_{12} & C_{11} & C_{12} & 0 & 0 & 0 \\ C_{12} & C_{12} & C_{11} & 0 & 0 & 0 \\ 0 & 0 & 0 & C_{44} & 0 & 0 \\ 0 & 0 & 0 & 0 & C_{44} & 0 \\ 0 & 0 & 0 & 0 & 0 & C_{44} \end{pmatrix}$ |
| Tetragonal | 6 | $C = \begin{pmatrix} C_{11} & C_{12} & C_{13} & 0 & 0 & 0 \\ C_{12} & C_{11} & C_{13} & 0 & 0 & 0 \\ C_{13} & C_{13} & C_{33} & 0 & 0 & 0 \\ 0 & 0 & 0 & C_{44} & 0 & 0 \\ 0 & 0 & 0 & 0 & C_{44} & 0 \\ 0 & 0 & 0 & 0 & 0 & C_{66} \end{pmatrix}$ |
| Orthorhombic | 9 | $C = \begin{pmatrix} C_{11} & C_{12} & C_{13} & 0 & 0 & 0 \\ C_{12} & C_{22} & C_{23} & 0 & 0 & 0 \\ C_{13} & C_{23} & C_{33} & 0 & 0 & 0 \\ 0 & 0 & 0 & C_{44} & 0 & 0 \\ 0 & 0 & 0 & 0 & C_{55} & 0 \\ 0 & 0 & 0 & 0 & 0 & C_{66} \end{pmatrix}$ |
| Triclinic | 21 | $C = \begin{pmatrix} C_{11} & C_{12} & C_{13} & C_{14} & C_{15} & C_{16} \\ C_{12} & C_{22} & C_{23} & C_{24} & C_{25} & C_{26} \\ C_{13} & C_{23} & C_{33} & C_{34} & C_{35} & C_{36} \\ C_{14} & C_{24} & C_{34} & C_{44} & C_{45} & C_{46} \\ C_{15} & C_{25} & C_{35} & C_{45} & C_{55} & C_{56} \\ C_{16} & C_{26} & C_{36} & C_{46} & C_{56} & C_{66} \end{pmatrix}$ |

Table 12: Number of independent components in the elastic tensor for different crystal systems.

| Crystal system | point groups | Number of independent elements | Piezoelectric tensor |
|---|---|---|---|
| Trigonal | 32 | 2 | $\mathbf{e} = \begin{pmatrix} \mathbf{e}_{11} & -\mathbf{e}_{11} & 0 & \mathbf{e}_{14} & 0 & 0 \\ 0 & 0 & 0 & 0 & -\mathbf{e}_{14} & -\mathbf{e}_{11} \\ 0 & 0 & 0 & 0 & 0 & 0 \end{pmatrix}$ |
| Monoclinic | 2 | 8 | $\mathbf{e} = \begin{pmatrix} 0 & 0 & 0 & \mathbf{e}_{14} & 0 & \mathbf{e}_{16} \\ \mathbf{e}_{21} & \mathbf{e}_{22} & \mathbf{e}_{23} & 0 & e_{25} & 0 \\ 0 & 0 & 0 & \mathbf{e}_{34} & 0 & \mathbf{e}_{36} \end{pmatrix}$ |
| Triclinic | 1 | 18 | $\mathbf{e} = \begin{pmatrix} \mathbf{e}_{11} & \mathbf{e}_{12} & \mathbf{e}_{13} & \mathbf{e}_{14} & \mathbf{e}_{15} & \mathbf{e}_{16} \\ \mathbf{e}_{21} & \mathbf{e}_{22} & \mathbf{e}_{23} & \mathbf{e}_{24} & \mathbf{e}_{25} & \mathbf{e}_{26} \\ \mathbf{e}_{31} & \mathbf{e}_{32} & \mathbf{e}_{33} & \mathbf{e}_{34} & \mathbf{e}_{35} & \mathbf{e}_{36} \end{pmatrix}$ |

Table 13: Number of independent components in the piezoelectric tensor for different crystal systems.

dielectric and piezoelectric tensor prediction task, the 512-dimensional vector is mapped to a 128-dimensional vector through a non-linear layer, while in the elastic tensor prediction, it is mapped to a 256-dimensional vector. For the prediction module, eComFormer, in the dielectric tensor prediction task, we used 4 node-wise transformer layers and 1 node-wise equivariant updating layer, whereas in the piezoelectric and elastic tensor prediction, we used 2 node-wise transformer layers and 1 node-wise equivariant updating layer. For both dielectric, piezoelectric, and elastic tensor tasks, the learning rate was set to 0.001, with 200 epochs and a batch size of 64. In the Reverse R&R module, for the dielectric tensor prediction task, the 128-dimensional vector features output by the prediction module are mapped to a 6-dimensional vector through two non-linear layers, then reconstructed into a $3 \times 3$ matrix and combined with the orthogonal matrix $\mathbf{Q}$ obtained from the R&R module to achieve $O(3)$ equivariant output. For the piezoelectric prediction task, the 18-dimensional vector features output by the prediction module are reconstructed into a $3 \times 6$ matrix and combined with the orthogonal matrix $\mathbf{Q}$ from the R&R module to achieve $O(3)$ equivariant output. For the elastic tensor prediction task, the 256-dimensional vector features output by the prediction module are mapped to a 36-dimensional vector through two non-linear layers, then reconstructed into a $6 \times 6$ matrix and combined with the orthogonal matrix $\mathbf{Q}$ from the R&R module to achieve $O(3)$ equivariant output.

**Hyperparameter settings of GMTNet, ETGNN, and MEGNET.** Following Yan et al. (2024b), we trained GMTNet, ETGNN, and MEGNET for 200 epochs using Huber loss with a learning rate of 0.001 and Adam optimizer with $10^{-5}$ weight decay across all tasks. The same polynomial learning rate decay scheduler is used in all experiments.

|  | 1st | 2nd | 3rd | 4th | 5th | mean |
|---|---|---|---|---|---|---|
| MEGNET |  |  |  |  |  |  |
| Fnorm ↓ | 3.62 | 3.70 | 3.66 | 3.74 | 3.81 | 3.71 |
| EwT 25% ↑ | 76.0% | 77.1% | 75.6% | 73.7% | 76.4% | 75.8 % |
| EwT 10% ↑ | 41.2% | 39.3% | 38.2% | 39.7% | 36.3% | 38.9% |
| EwT 5% ↑ | 17.6% | 16.8% | 18.5% | 19.1% | 17.8% | 18.0% |
| ETGNN |  |  |  |  |  |  |
| Fnorm ↓ | 3.42 | 3.48 | 3.33 | 3.35 | 3.41 | 3.40 |
| EwT 25% ↑ | 82.1% | 81.7% | 83.4% | 82.6% | 83.2% | 82.6 % |
| EwT 10% ↑ | 46.9% | 47.8% | 50.7% | 49.4% | 50.5% | 49.1% |
| EwT 5% ↑ | 22.3% | 26.5% | 27.1% | 25.9% | 24.6% | 25.3% |
| GMTNet |  |  |  |  |  |  |
| Fnorm ↓ | 2.98 | 3.64 | 3.23 | 3.16 | 3.39 | 3.28 |
| EwT 25% ↑ | 83.4% | 81.3% | 83.8% | 84.5% | 83.7% | 83.3 % |
| EwT 10% ↑ | 54.9% | 56.2% | 55.8% | 56.2% | 57.1% | 56.0% |
| EwT 5% ↑ | 31.2% | 30.5% | 28.8% | 29.3% | 32.7% | 30.5% |
| **GoeCTP (Ours)** |  |  |  |  |  |  |
| Fnorm ↓ | 3.33 | 3.29 | 3.36 | 3.25 | 2.92 | 3.23 |
| EwT 25% ↑ | 82.5% | 83.4% | 82.1% | 83.9% | 84.2% | 83.2 % |
| EwT 10% ↑ | 57.5% | 53.9% | 56.4% | 56.7% | 59.7% | 56.8% |
| EwT 5% ↑ | 36.9% | 31.6% | 37.1% | 34.2% | 37.8% | 35.5% |

Table 14: Comparison of 5 repeated experiments on dielectric dataset.

|  | MEGNET | ETGNN | GMTNet | **GoeCTP (eCom.)** | **GoeCTP (iCom.)** | **GoeCTP (Crys.)** |
|---|---|---|---|---|---|---|
| Fnorm ↓ | 3.71 | 3.40 | 3.28 | **3.23** | 3.40 | 3.53 |
| EwT 25% ↑ | 75.8% | 82.6% | **83.3%** | 83.2% | 81.7% | 80.1% |
| EwT 10% ↑ | 38.9% | 49.1% | 56.0% | **56.8%** | 53.8% | 52.9% |
| EwT 5% ↑ | 18.0% | 25.3% | 30.5% | **35.5%** | 32.3% | 30.6% |
| Total Time (s) ↓ | 663 | 1325 | 1611 | 616 | **535** | 645 |
| Time/batch (s) ↓ | 0.052 | 0.104 | 0.126 | 0.048 | **0.042** | 0.202 |

Table 15: Additional comparison of performance metrics on dielectric dataset.

**Additional results.** Given the relatively small size of the dielectric tensor dataset, along with other influencing factors, we observed some randomness in the results during repeated training runs, introducing noise into the outcome comparisons. To mitigate this effect and ensure the reliability of the results, we conducted five independent trials for each method, using the average of the performance metrics across these five trials as the final reported value. The detailed results from these repeated experiments are presented in Table 14.

We assessed the performance of GoeCTP when combined with iComFormer (Yan et al., 2024a) and CrystalFormer (Taniai et al., 2024) (denoted as GoeCTP (iCom.) for the combination) on both the dielectric tensor and elastic tensor datasets. As presented in Table 15, on the dielectric tensor dataset, although the prediction accuracy of GoeCTP (iCom.) and GoeCTP (Crys.) did not exceed that of GoeCTP (eCom.), GoeCTP (iCom.) demonstrated slightly higher computational efficiency compared to GoeCTP (eCom.). In contrast, as shown in Table 16, on the elastic tensor dataset, GoeCTP (iCom.) and GoeCTP (Crys.) achieved superior prediction quality relative to GoeCTP (eCom.), albeit with lower efficiency.

## A.5 THE EXTENSIBILITY AND LIMITATIONS OF OUR METHOD

**limitations.** (1)As discussed in the main text, GoeCTP is a plug-and-play $O(3)$-equivariant framework designed to enhance the backbone network's ability to achieve equivariant predictions. Consequently, the performance of our method is inherently dependent on the capabilities of the backbone

| | MEGNET | ETGNN | GMTNet | GoeCTP (eCom.) | GoeCTP (iCom.) | GoeCTP (Crys.) |
|---|---|---|---|---|---|---|
| Fnorm ↓ | 143.86 | 123.64 | 117.62 | 107.11 | **102.80** | 107.44 |
| EwT 25% ↑ | 23.6% | 32.0% | 36.0% | 42.5% | **46.7%** | 43.5% |
| EwT 10% ↑ | 3.0% | 3.8% | 7.6% | 15.3% | **18.6%** | 15.8% |
| EwT 5% ↑ | 0.5% | 0.5% | 2.0% | 7.2% | **8.2%** | 7.9% |
| Total Time (s) ↓ | 2899 | 4448 | > 36000 | **2422** | 4035 | 7891 |
| Time/batch (s) ↓ | 0.226 | 0.348 | > 2.813 | **0.189** | 0.315 | 0.616 |

Table 16: Additional comparison of performance metrics on elastic dataset.

network. If the backbone lacks sufficient predictive power, the combined framework may not fully surpass the state-of-the-art tensor prediction networks in all evaluation metrics. For instance, as shown in Table 15, when our framework is integrated with CrystalFormer (Taniai et al., 2024), although the computational speed significantly exceeds that of GMTNet, the Fnorm, EwT 25%, and EwT 10% values are lower than those achieved by GMTNet.

(2)As demonstrated in Higham (1986) and Proposition 3.1, polar decomposition applied to a $3 \times 3$ invertible matrix $\mathbf{L}$ produces a unique $3 \times 3$ orthogonal matrix $\mathbf{Q}$. For 3D crystal structures, the lattice matrix $\mathbf{L}$ is always full-rank (i.e. invertible), therefore, ensuring the applicability of our method to 3D crystal systems. However, for certain special cases, such as 2D crystals with single-layer structures (Novoselov et al., 2005; Sherrell et al., 2022), the rank of the lattice matrix $\mathbf{L}$ may be less than 3. In these scenarios, directly applying polar decomposition may not yield an unique $3 \times 3$ orthogonal matrix $\mathbf{Q}$ and an unique standardized crystal position. $\mathbf{H}$. This limitation could cause our method to fail, as crystals with different spatial orientations cannot be consistently adjusted to a unique standardized position, thereby preventing the method from achieving the desired $O(3)$-equivariance.

**Extensibility.** For the extensibility of our proposed method, a natural extension would be to adapt it for use with 3D point cloud systems, such as 3D molecular systems. The ability of our method to directly transmit rotational information renders it particularly effective for specialized equivariant tasks. For instance, in equivariant 3D molecular generation tasks, our approach can directly transfer equivariant information from time step 0 to time step $T$, bypassing multiple denoising networks (Hoogeboom et al., 2022). This is previous equivariant techniques such as equivariant network (Satorras et al., 2021) and frame averaging (Puny et al., 2022; Lin et al., 2024) cannot achieve. Specifically, a molecule in 3D space can be represented as $\mathbf{M} = (\mathbf{A}, \mathbf{X})$, where $\mathbf{A} \in \mathbb{R}^{d_a \times n}$ denotes $d_a$-dimensional atom features for $n$ atoms in the molecular, and $\mathbf{X} \in \mathbb{R}^{3 \times n}$ contains the 3D coordinates of these $n$ atoms. To extend our method, polar decomposition would be applied to the coordinate matrix, such that $\mathbf{X} = \mathbf{Q} \exp(\mathbf{S})$. Then, the rotational information $\mathbf{Q}$ can be transferred to the output of the network to complete the equivariant task. As demonstrated in Higham (1986), if the coordinate matrix is full-rank, the polar decomposition is unique, enabling a seamless extension of our method to 3D molecular systems. However, there are special cases where this approach may fail. For example, in a molecule composed of three atoms all lying within the same plane, the coordinate matrix $\mathbf{X}$ may not be full-rank (e.g., when all atomic coordinates lie on a plane passing through the origin). This is similar to the scenario where our method is applied to 2D crystals, where directly applying polar decomposition may not yield a unique $3 \times 3$ orthogonal matrix $\mathbf{Q}$ and a unique standardized position $\mathbf{H}$. Additionally, since proposed framework operates externally to the network and does not participate in actual training, it may potentially assist large language models in achieving equivariant prediction tasks.

### A.6 HOW TO FURTHER UTILIZE THE TENSOR PROPERTIES SYMMETRY

In this section, we will use the dielectric tensor as an example to briefly discuss how to further utilize the symmetry of tensor properties.

**Utilizing symmetry constraints for zero elements.** We first present an example of GoeCTP prediction results, as shown in Fig. 17. It can be observed that, within a certain margin of error, our results are relatively consistent with the constraints.

| Label | Prediction | Cubic dielectric tensor |
|---|---|---|
| $\begin{pmatrix} 2.258 & 0 & 0 \\ 0 & 2.258 & 0 \\ 0 & 0 & 2.258 \end{pmatrix}$ | $\begin{pmatrix} 2.252 & 0.016 & 0.008 \\ 0.016 & 2.230 & 0.007 \\ 0.008 & 0.007 & 2.262 \end{pmatrix}$ | $\varepsilon = \begin{pmatrix} \varepsilon_{11} & 0 & 0 \\ 0 & \varepsilon_{11} & 0 \\ 0 & 0 & \varepsilon_{11} \end{pmatrix}$ |

Table 17: An example of GoeCTP prediction results

For a dielectric tensor, using 1% of the average value of non-zero elements in the labels as a threshold, we judge whether the prediction for a zero element was successful. The results are as shown in Fig. 18.

| Crystal system | Cubic | Tetragonal | Hexagonal-Trigonal | Orthorhombic | Monoclinic |
|---|---|---|---|---|---|
| Success rate | 88.3% | 86.6% | 84.5% | 84.5% | 75.7% |

Table 18: The GoeCTP results of predicting symmetry-constrained zero-valued dielectric tensor elements.

It can be observed that our method successfully predicts most zero elements, but it is not perfect. Since our advantage lies in transferring equivariance through an external framework, there are no restrictions on the model itself. Therefore, to achieve a higher success rate for zero elements, we added a ReLU activation function to the output layer of the network to improve the success rate (this applies only to cases where tensor elements are greater than or equal to zero; for other cases, specific activation functions need to be designed, such as $ReLU(x - 0.01) - ReLU(-x)$). The results after retraining GeoCTP are as follows:

| Label | Prediction | Cubic dielectric tensor |
|---|---|---|
| $\begin{pmatrix} 2.258 & 0 & 0 \\ 0 & 2.258 & 0 \\ 0 & 0 & 2.258 \end{pmatrix}$ | $\begin{pmatrix} 2.237 & 0.000 & 0.000 \\ 0.000 & 2.283 & 0.000 \\ 0.000 & 0.000 & 2.228 \end{pmatrix}$ | $\varepsilon = \begin{pmatrix} \varepsilon_{11} & 0 & 0 \\ 0 & \varepsilon_{11} & 0 \\ 0 & 0 & \varepsilon_{11} \end{pmatrix}$ |

Table 19: An example of GoeCTP (ReLU) prediction results

| Crystal system | Cubic | Tetragonal | Hexagonal-Trigonal | Orthorhombic | Monoclinic |
|---|---|---|---|---|---|
| Success rate | 100% | 100% | 87.2% | 100% | 100% |

Table 20: The GoeCTP (ReLU) results of predicting symmetry-constrained zero-valued dielectric tensor elements.

| | GoeCTP | GoeCTP(relu) |
|---|---|---|
| Fnorm ↓ | 3.23 | 3.26 |
| EwT 25% ↑ | 83.2% | 82.6% |
| EwT 10% ↑ | 56.8% | 58.4% |
| EwT 5% ↑ | 35.5% | 36.3% |

Table 21: Predictive performance comparisons between GoeCTP and GoeCTP (ReLU) on the dielectric dataset.

This simple modification allows our method to more accurately predict the zero elements in dielectric tensors caused by the space group.

**Utilizing symmetry constraints for non-zero elements.** The example of utilizing symmetry constraints for zero elements is a simple illustration; when the space group of the input crystal is known, the prior knowledge in Appendix A.3 can be used to ensure that the network output fully obeys the constraints of tensor properties. For instance, the mask from Table 11 can be applied to weight

the network output, ensuring 100% compliance with the constraints. This was not experimentally demonstrated in our current work, but we plan to explore related studies in future work.

