# OpenReview forum: "Fast Crystal Tensor Property Prediction: A General O(3)-Equivariant Framework Based on Polar Decomposition"
_ICLR.cc/2025/Conference — Submitted to ICLR 2025_

### Official Review · Reviewer_pK6i · 2024-10-17

**Soundness:** 3
**Presentation:** 3
**Contribution:** 2
**Rating:** 6
**Confidence:** 3

**Summary:**

The paper introduces a method for predicting how crystals react to forces applied from different directions, a challenge that requires maintaining consistency regardless of the crystal's orientation in space.

The proposed method uses a sound mathematical technique (polar decomposition) to standardize crystal positions, enabling faster and more accurate predictions of these directional properties (known technically as tensor properties) while respecting the physics principle of orientation independence (technically called O(3) group equivariance).

The proposal is significantly faster and more accurate than existing approaches, especially for predicting how materials deform under stress or respond to electric fields.

**Strengths:**

1. The pre-processing step of standardizing crystal positions through polar decomposition is significant for multiple reasons. It ensures equivariance, simplifies model architecture, and preserves tensor properties across different orientations through an additional step.
2. The method achieves 13x speed improvement over prior methods in predicting tensor properties.
3. The paper effectively explains O(3)-equivariance in crystal tensor prediction through clear organisation, helpful diagrams, and accessible mathematical explanations, making sophisticated concepts understandable even to non-specialists.

**Weaknesses:**

1. The paper focuses primarily on quantitative metrics (e.g. Frobenius norm and EwT percentages) to demonstrate the method's effectiveness. However, there is a lack of qualitative insights into how the method affects real-world predictions. Including case studies or qualitative analyses where the method's predictions are compared to known physical properties of materials would strengthen the practical significance.
2. Evaluation is carried out on two specific datasets for dielectric and elastic tensor prediction. While the results are promising, these datasets may not cover the full range of tensor property prediction challenges. Testing on a broader variety of materials, including more extreme cases, would strengthen the claim of generalisability.
3. As the authors discuss between lines 137 and 142, " the requirements for O(3) equivariance typically differ from the O(3)-equivariance
defined in the general molecular studies." Because of these specific requirements, the scope and the applicability of the proposed polar decomposition are limited.

**Questions:**

1. Were there qualitative case studies where the proposed method's predictions were compared to known real-world material properties, such as elastic or dielectric responses of specific materials?
2. Were there plans to test the proposed method on other datasets, especially those involving more complex or extreme tensor property cases (e.g., materials with highly anisotropic properties or rare crystal structures)?
3. Why was the dataset "Piezo" that was tested in a prior work [Yan et al.] not tested in this paper?
4. What explains the discrepancy in the number of samples in the "Elastic" dataset between the prior work [Yan et al.] and this paper? The prior work [Yan et al.] reports 14,220 samples in Table 1, while this paper reports 25,110 samples in Table 1 on line 364.
5. Could the specific requirements of O(3) equivariance for crystalline materials limit the use of polar decomposition? Additionally, how did these differences impact the potential generalisation of the proposed method to molecular systems, where O(3) equivariance is defined differently?

[Yan et al.] [A Space Group Symmetry Informed Network for O(3) Equivariant Crystal Tensor Prediction, In ICML 2024](https://openreview.net/forum?id=BOFjRnJ9mX).

---

> ### Author Response · Authors · 2024-11-22
>
> We sincerely thank you for your recognition of our method in terms of speed improvement and clear organisation. Below we will
> address your questions in detail.
>
> ---------------
>
> **$\bullet$  Response to weakness 1 and question 1**
>
>
> Thank you for the comments and suggestions. The purpose of deep learning (DL) methods used in property prediction is to replace slow, DFT-like physical simulation methods, thereby accelerating the speed of property predictions.
> The datasets used to train DL methods are also derived from DFT-like physical simulations. As a result, current deep learning prediction methods aim to achieve prediction accuracy close to DFT-like simulations. Therefore, following previous work [1], our research does not involve specific real-world materials.
> In [2], comparisons between DFT and real materials are discussed. In the future, we will attempt similar comparisons with real-world materials as in [2].
> We hope our response satisfies the reviewer.
>
> [1] Yan, Keqiang and Saxton, Alexandra and Qian, Xiaofeng and Qian, Xiaoning and Ji, Shuiwang. A Space Group Symmetry Informed Network for O (3) Equivariant Crystal Tensor Prediction. *ICML2024*.
>
> [2] Petousis, Ioannis and Chen, Wei and Hautier, Geoffroy and Graf, Tanja and Schladt, Thomas D and Persson, Kristin A and Prinz, Fritz B. Benchmarking density functional perturbation theory to enable high-throughput screening of materials for dielectric constant and refractive index. *Physical Review B*, 2016, 93(11): 115151.
>
> --------------
>
> **$\bullet$  Response to weakness 2 and question 2**
>
> The publicly available datasets we have collected so far mainly involve dielectric tensors, elastic tensors, and piezoelectric tensors. For this rebuttal, we have added experiments on the piezoelectric tensor dataset. If we discover other publicly available datasets in the future, we will continue testing the proposed method.
>
>
> ------------------
>
> **$\bullet$  Response to Weaknesses 3 and question 5**
>
>
> We thank the reviewer for this helpful comment.
> We include additional limitation discussions and potential generalization of the proposed method to molecular systems in the Appendix A.5.
> Specifically, for certain special cases, such as 2D crystals with single-layer structures [1,2], the rank of the lattice matrix $\mathbf{L}$ may be less than 3. Therefore, directly applying polar decomposition may not yield a unique $3 \times 3$ orthogonal matrix $\mathbf{Q}$ [3]. This limitation could cause our method to fail to achieve the desired $O(3)$-equivariance.
> When our method is applied to certain 2D molecules in 3D space (for example, a molecule composed of three atoms all lying within the same plane), it also shares similar limitations to those encountered with 2D crystals.
> We hope our response addresses the reviewer’s question.
>
>
> [1] Novoselov, Kostya S and Jiang, Da and Schedin, F and Booth, TJ and Khotkevich, VV and Morozov, SV and Geim, Andre K. Two-dimensional atomic crystals. *Proceedings of the National Academy of Sciences*, 2005, 102(30): 10451-10453.
>
> [2] Sherrell, Peter C and Fronzi, Marco and Shepelin, Nick A and Corletto, Alexander and Winkler, David A and Ford, Mike and Shapter, Joseph G and Ellis, Amanda V. A bright future for engineering piezoelectric 2D crystals. *Chemical Society Reviews*, 2022, 51(2): 650-671.
>
> [3] Higham N J. Computing the polar decomposition—with applications. *SIAM Journal on Scientific and Statistical Computing*, 1986, 7(4): 1160-1174.
>
>
> --------------
>
>
> **$\bullet$  Response to question 3**
>
> We have added additional experiments using the piezoelectric tensor dataset, which is from ref. [1]. The experimental results are as follows:
>
> |                          | MEGNET  | ETGNN   | GMTNet  | **GoeCTP (Ours)** |
> |--------------------------|---------|---------|---------|------------------|
> | **Fnorm ↓**              | 0.465   | 0.535   | 0.462   | **0.448**        |
> | **EwT 25% ↑**            | 43.9%   | 37.5%   | **45.7%** | 44.9%            |
> | **EwT 10% ↑**            | 37.9%   | 22.8%   | 39.3%   | **43.1%**        |
> | **EwT 5% ↑**             | 27.1%   | 13.8%   | 35.7%   | **40.1%**        |
>
> *Table: Predictive performance comparisons among MEGNET, ETGNN, GMTNet, and GoeCTP on the piezoelectric dataset.*
>
>
> Our method achieves optimal results on Fnorm, EwT 5\%, and EwT 10\% metrics.
>  We hope our response addresses the reviewer's concerns.
>
>
> [1] Yan, Keqiang and Saxton, Alexandra and Qian, Xiaofeng and Qian, Xiaoning and Ji, Shuiwang. A Space Group Symmetry Informed Network for O (3) Equivariant Crystal Tensor Prediction. *ICML2024*.

---

> > ### Author Response · Authors · 2024-11-22
> >
> > -----------------------
> >
> > **$\bullet$  Response to question 4**
> >
> >
> > We thank the reviewer for the astute observation. Before the ICLR submission deadline, the authors of GMTNet had not released their collected Elastic Tensors dataset. Therefore, we evaluated our collected Elastic Tensors dataset (from dft\_3d in https://pages.nist.gov/jarvis/databases/), introducing the performance discrepancy. In response to the reviewer's concern, we reevaluated the performance with the datasets used by GMTNet (released on Nov. 5), and the results are as follows:
> >
> >
> > |                          | GMTNet       | **GoeCTP (Ours)** |
> > |--------------------------|--------------|-------------------|
> > | **Fnorm ↓**              | 80.06        | 72.43            |
> > | **EwT 25% ↑**            | 57.8%        | 62.2%            |
> > | **EwT 10% ↑**            | 14.9%        | 27.5%            |
> > | **EwT 5% ↑**             | 4.6%         | 14.7%            |
> > | **Total Time (s) ↓**     | >24000       | 1815.55          |
> >
> > *Table: Predictive performance comparisons between GMTNet and GoeCTP on another elastic dataset.*
> >
> >
> > We can see that our method still outperforms GMTNet.

---

### Official Review · Reviewer_3tQt · 2024-10-28

**Soundness:** 2
**Presentation:** 2
**Contribution:** 2
**Rating:** 3
**Confidence:** 4

**Summary:**

This paper proposes a novel approach intended to achieve O(3) tensorial equivariance by transforming crystal structures into standardized positions, so that neural networks do not need to satisfy equivariance during prediction. The invariant predictions are then mapped back to equivariant outputs. While the goal of O(3) equivariant predictions is notable, prior work has already addressed this problem with established solutions, including ETGNN's vector outer product and GMTNet's equivariant networks. Additionally, existing techniques, such as frame averaging and minimal frame averaging, can be employed to achieve O(3) tensor equivariance effectively. Moreover, this work does not consider space group constraints, which are crucial for tensorial properties in crystallography. As such, the novelty and contribution of this work are limited and do not meet the standards for acceptance at current form.

**Strengths:**

1. An alternative approach is proposed to achieve tensor O(3) equivariance. If used, it is faster than equivariant network based O(3) equivariant predictions like GMTNet.

**Weaknesses:**

1. **Lack of Consideration for Space Group Constraints**

Space group constraints, which are fundamental in determining the tensor properties of crystals, are not accounted for in this work. No experimental results are provided to verify whether the proposed method can generate predictions that align with these constraints. Crystals exhibit unique tensor characteristics that are intrinsically tied to their crystal class or space group, and ignoring these symmetries is a significant oversight.

2. **Limited Improvement in Performance**

The integration of the proposed module does not enhance eComformer’s performance beyond achieving O(3) equivariance, as shown in Table 4. Other alternatives, such as ETGNN, frame averaging, and minimal frame averaging, also achieve O(3) equivariance but were not discussed or compared in this work. A more thorough discussion and comparative analysis of technical contributions and novelty would be beneficial.

3. **Absence of Experiments on Piezoelectric Tensors**

The work lacks experiments on piezoelectric tensors, which are especially sensitive to space group constraints. Including these experiments would strengthen the evaluation of the proposed approach’s applicability across tensorial properties with varying sensitivity to symmetry constraints.

4. **Performance on Elastic Tensors**

The performance of the proposed method on elastic tensors is significantly lower than GMTNet's original results. This suggests potential limitations in the approach’s effectiveness.

5. **Efficiency Gains Not Attributable to Proposed Method**

The efficiency gains claimed largely derive from the use of the lightweight eComformer, not the proposed approach. Similar speed-ups could be achieved by combining eComformer with other O(3) equivariant methods such as ETGNN’s vector outer-product approach or minimal frame averaging.

**Questions:**

As listed above in Weaknesses.

---

> ### Author Response · Authors · 2024-11-22
>
> We sincerely thank you for your valuable time and comments.
> Below we will address your concerns in detail.
>
> --------------
>
> **$\bullet$  Response to weakness 1**
>
>
> Specifically, our method is space group invariant. According to [1] and [2], space group transformations do not change the lattice matrix $L$, i.e., $RL+b=L$. Therefore, when space group transformations are applied to a crystal, applying polar decomposition yields the same result, i.e., $RL+b=L=Q\exp(S)$. Thus, when space group transformations are applied to the input, our output remains invariant.
>
> Additionally, we have discussed the space group constraints in Appendix A.3. We acknowledge that our current utilization of such physical knowledge is limited. Further leveraging space group constraints to improve tensor prediction performance is a direction for our future research (For example, predicting only the independent components of tensor properties to improve prediction performance).
>
> [1] Jiao, Rui and Huang, Wenbing and Liu, Yu and Zhao, Deli and Liu, Yang. Space Group Constrained Crystal Generation. *ICLR2024*.
>
> [2] Yan, Keqiang and Saxton, Alexandra and Qian, Xiaofeng and Qian, Xiaoning and Ji, Shuiwang. A Space Group Symmetry Informed Network for O (3) Equivariant Crystal Tensor Prediction. *ICML2024*.
>
>
>
> -----------------------
>
>
> **$\bullet$  Response to weakness 2 and weakness 5**
>
>
> In response to your comment, we first conducted experiments combining ETGNN’s vector outer-product approach and SOTA Minimal Frame Averaging [1] with eComFormer. The results are as follows:
>
> |                              | eCom. (vector outer-product) | eCom. (Minimal Frame Averaging) | **GoeCTP (Ours)** |
> |------------------------------|------------------------------|----------------------------------|-------------------|
> | **Fnorm ↓**                 | 3.78                        | **3.20**                        | 3.23              |
> | **EwT 25% ↑**               | 76.4%                       | **83.5%**                       | 83.2%             |
> | **EwT 10% ↑**               | 32.5%                       | 56.0%                           | **56.8%**         |
> | **EwT 5% ↑**                | 14.0%                       | 32.4%                           | **35.5%**         |
> | **Total Time (s) ↓**        | 1078                        | 639                              | **616**           |
>
> *Table: Predictive performance comparisons between eComFormer (vector outer-product), eComFormer (Minimal Frame Averaging), and GoeCTP on the dielectric dataset.*
>
>
> Based on results, it is clear that our method  outperforms eComFormer with
> ETGNN’s vector outer-product approach in all metrics, and our method also outperforms Minimal Frame Averaging in terms of prediction quality, i.e. EwT 5\%.
>
>
>
> 1) **Comparison for ETGNN’s vector outer-product.** ETGNN’s vector outer-product requires a specially designed network structure, involving the introduction of tensor products to achieve $O(3)$-equivariance. This adds extra computational cost to the network. Moreover, since $O(3)$-equivariance relies on tensor products, it is not possible to apply weighting to each position in the $3 \times 3$ tensor product result, which limits the effectiveness of ETGNN’s vector outer-product. In contrast, our method is independent of the network structure and, therefore, does not face these issues.
>
> 2) **Comparison for frame averaging [1,2].**
> First, directly applying frame averaging methods cannot simultaneously achieve both space group invariance and $O(3)$ equivariance for crystals. Without considering translations, space groups are the subgroup of the $O(3)$ group. Achieving both $O(3)$ equivariance and invariance to $O(3)$ subgroup through frame averaging methods is inherently contradictory. Furthermore, as demonstrated in Ref. [3], using frame averaging to ensure space group invariance often degrades prediction performance. In contrast, our method leverages the lattice matrix and polar decomposition to directly achieve space group invariance at the data representation level. This process is independent of the subsequent $O(3)$ equivariance framework. By decoupling these two processes, our approach avoids conflicts and simultaneously achieves both objectives effectively.
> Second, frame averaging methods involve modifying the loss function, increasing its complexity, which in turn raises the training  computational cost of the network and reduces efficiency. In contrast, as an external framework, our method imposes no additional computational burden on the network itself.

---

> > ### Author Response · Authors · 2024-11-22
> >
> > 3) **For Table 4.**
> > According to our description of Table 4, both the first and third columns correspond to the results of GoeCTP. Our goal is to compare the differences between the results in the first and second columns and those in the third and fourth columns, which demonstrate how much the performance of eComFormer degrades without our framework.
> > In fact, the results of directly applying eComFormer to tensor prediction tasks are as follows:
> >
> >
> > |                          | eComFormer | **GoeCTP (Ours)** |
> > |--------------------------|------------|-------------------|
> > | **Fnorm ↓**              | 3.60       | 3.23             |
> > | **EwT 25% ↑**            | 80.6%      | 83.2%            |
> > | **EwT 10% ↑**            | 56.2%      | 56.8%            |
> > | **EwT 5% ↑**             | 32.6%      | 35.5%            |
> > | **Total Time (s) ↓**     | 613        | 616              |
> >
> > *Table: Predictive performance comparisons between eComFormer and GoeCTP on the dielectric dataset.*
> >
> >
> > We can see that our method outperforms eComFormer.
> >
> >
> >
> > [1] Lin Y, Helwig J, Gui S, Ji S. Equivariance via Minimal Frame Averaging for More Symmetries and Efficiency. *ICML2024*.
> >
> >
> > [2] Omri Puny, Matan Atzmon, Edward J Smith, Ishan Misra, Aditya Grover, Heli Ben-Hamu, and
> > Yaron Lipman. Frame averaging for invariant and equivariant network design. *ICLR2022*.
> >
> >
> > [3] Yan, Keqiang and Saxton, Alexandra and Qian, Xiaofeng and Qian, Xiaoning and Ji, Shuiwang. A Space Group Symmetry Informed Network for O (3) Equivariant Crystal Tensor Prediction. *ICML2024*.
> >
> > -------------
> >
> >
> > **$\bullet$  Response to weakness 3**
> >
> >
> >
> >
> > In response to weaknesses 3, we have added additional experiments on Piezoelectric tensor dataset, which is from ref.[1]. The experimental results are as follows:
> >
> > |                          | MEGNET  | ETGNN   | GMTNet  | **GoeCTP (Ours)** |
> > |--------------------------|---------|---------|---------|------------------|
> > | **Fnorm ↓**              | 0.465   | 0.535   | 0.462   | **0.448**        |
> > | **EwT 25% ↑**            | 43.9%   | 37.5%   | **45.7%** | 44.9%            |
> > | **EwT 10% ↑**            | 37.9%   | 22.8%   | 39.3%   | **43.1%**        |
> > | **EwT 5% ↑**             | 27.1%   | 13.8%   | 35.7%   | **40.1%**        |
> >
> > *Table: Predictive performance comparisons among MEGNET, ETGNN, GMTNet, and GoeCTP on the piezoelectric dataset.*
> >
> >
> > Our method achieves optimal results on Fnorm, EwT 5\%, and EwT 10\% metrics.
> >
> >
> > [1] Yan, Keqiang and Saxton, Alexandra and Qian, Xiaofeng and Qian, Xiaoning and Ji, Shuiwang. A Space Group Symmetry Informed Network for O (3) Equivariant Crystal Tensor Prediction. *ICML2024*.
> >
> >
> >
> > **$\bullet$  Response to weakness 4**
> >
> >
> > We thank the reviewer for the astute observation. Before the ICLR submission deadline, the authors of GMTNet had not released their collected Elastic tensors dataset. Therefore, we evaluated with our collected Elastic Tensors dataset (from dft\_3d in https://pages.nist.gov/jarvis/databases/). Evaluating different datasets introduces the performance discrepancy. In response to the reviewer's concern, we re-evaluated the performance with the datasets used by GMTNet (released on Nov. 5), and the results are as follows:
> >
> >
> > |                          | GMTNet       | **GoeCTP (Ours)** |
> > |--------------------------|--------------|-------------------|
> > | **Fnorm ↓**              | 80.06        | 72.43            |
> > | **EwT 25% ↑**            | 57.8%        | 62.2%            |
> > | **EwT 10% ↑**            | 14.9%        | 27.5%            |
> > | **EwT 5% ↑**             | 4.6%         | 14.7%            |
> > | **Total Time (s) ↓**     | >24000       | 1815.55          |
> >
> > *Table: Predictive performance comparisons between GMTNet and GoeCTP on another elastic dataset.*
> >
> >
> > We can see that our method still outperforms GMTNet.

---

> > > ### Comment · Reviewer_3tQt · 2024-11-22
> > > **Thank you for your responses**
> > >
> > > Thank you for your extensive effort during the rebuttal process.
> > >
> > > Regarding the first question, the concern is not about space group invariance but about adhering to space group constraints for various crystal systems. For instance, if a method violates space group constraints and generates non-zero entries in off-diagonal positions of dielectric tensors for cubic systems, it would be incorrect. Predictions must satisfy space group constraints to be practically useful because these tensors describe the system's response to external fields or other perturbations. If the predictions do not respect these constraints, the responses will not align with the underlying crystal system's symmetry.
> > >
> > > For the second question, it appears that minimal frame averaging could achieve performance comparable to the proposed method, at least in terms of Fnorm and EwT 25 metrics.
> > >
> > > Regarding Table 4, the explanation remains unclear despite your comments. Specifically, what does it mean that the first column corresponds to the results of GoeCTP but is labeled as eComFormer?
> > >
> > > For weaknesses 3 and 4, if you are using the same dataset as GMTNet, the performance reported in the original GMTNet paper appears to be better than that of the proposed GoeCTP. For example, GMTNet achieves an Fnorm of 0.37 for piezoelectric tensors and 67 for elastic tensors, which outperform the results of GoeCTP.
> > >
> > > Thank you once again for your thorough and formal responses. While I appreciate the effort, the above concerns remain unresolved, and I cannot increase the score at this time.

---

> > > > ### Author Response · Authors · 2024-11-25
> > > > **Response to Reviewer 3tQt  (round2)**
> > > >
> > > > We sincerely thank you for reviewing again. We will attempt to address your concerns further.
> > > >
> > > > ------------------------
> > > >
> > > > **$\bullet$  Response to the first question (round2)**
> > > >
> > > > We have added relevant discussions in Appendix A.6 of the paper, as detailed below:
> > > >
> > > > **(1)** We first present an example of our prediction results, as shown below.
> > > >
> > > > | Label                                                                                          | Prediction                                                                                      | Cubic Dielectric Tensor                                                                                     |
> > > > |------------------------------------------------------------------------------------------------|------------------------------------------------------------------------------------------------|------------------------------------------------------------------------------------------------------------|
> > > > | $\begin{pmatrix}2.258&0&0\\\0&2.258&0\\\0&0&2.258\end{pmatrix}$                                 | $\begin{pmatrix}2.252&0.016&0.008\\\0.016&2.230&0.007\\\0.008&0.007&2.262\end{pmatrix}$             | $\begin{pmatrix} e&0&0\\\0&e&0\\\0&0&e\end{pmatrix}$ |
> > > >
> > > > *Table: An example of GoeCTP prediction results*
> > > >
> > > > It can be observed that, within a certain margin of error, our results are relatively consistent with the constraints.
> > > >
> > > > For a dielectric tensor, using 1\% of the average value of non-zero elements in the labels as a threshold, we judge whether the prediction for a zero element was successful. The results are as follows.
> > > >
> > > > | Crystal system      | Cubic | Tetragonal | Hexagonal-Trigonal | Orthorhombic | Monoclinic |
> > > > |---------------------|-------|------------|---------------------|--------------|------------|
> > > > | **Success rate**    | 88.3% | 86.6%      | 84.5%              | 84.5%        | 75.7%      |
> > > >
> > > > *Table: The GoeCTP results of predicting symmetry-constrained zero-valued dielectric tensor elements.*
> > > >
> > > > It can be seen that our method successfully predicts most zero elements.
> > > >
> > > > ------------------------
> > > >
> > > > **(2)** To get better success rate. Since our advantage lies in transferring equivariance through an external framework, there are no restrictions on the model. Therefore, we added a ReLU activation function to the network's output layer to improve  the success rate (This applies only to cases where the elements in the tensor are greater than or equal to zero; other cases require designing specific activation functions, such as $ReLU(x-0.01)-ReLU(-x)$.). After retraining, the results are as follows:
> > > >
> > > >
> > > > | Label                                                                                          | Prediction                                                                                      | Cubic Dielectric Tensor                                                                                     |
> > > > |------------------------------------------------------------------------------------------------|------------------------------------------------------------------------------------------------|------------------------------------------------------------------------------------------------------------|
> > > > | $\begin{pmatrix}2.258&0&0\\\0&2.258&0\\\0&0&2.258\end{pmatrix}$                                 | $\begin{pmatrix}2.237&0.000&0.000\\\0.000&2.283&0.000\\\0.000&0.000&2.228\end{pmatrix}$             | $\begin{pmatrix} e&0&0\\\0&e&0\\\0&0&e\end{pmatrix}$ |
> > > >
> > > > *Table: An example of GoeCTP (ReLU) prediction results*
> > > >
> > > > | Crystal system      | Cubic | Tetragonal | Hexagonal-Trigonal | Orthorhombic | Monoclinic |
> > > > |---------------------|-------|------------|---------------------|--------------|------------|
> > > > | **Success rate**    | 100%  | 100%       | 87.2%              | 100%         | 100%       |
> > > >
> > > > *Table: The GoeCTP (ReLU) results of predicting symmetry-constrained zero-valued dielectric tensor elements.*
> > > >
> > > > | Metric               | GoeCTP  | GoeCTP (ReLU) |
> > > > |----------------------|---------|---------------|
> > > > | **Fnorm ↓**         | 3.23    | 3.26          |
> > > > | **EwT 25% ↑**       | 83.2%   | 82.6%         |
> > > > | **EwT 10% ↑**       | 56.8%   | 58.4%         |
> > > > | **EwT 5% ↑**        | 35.5%   | 36.3%         |
> > > >
> > > > *Table: Predictive performance comparisons between GoeCTP and GoeCTP (ReLU) on the dielectric dataset.*
> > > >
> > > > This simple modification allows our method to more accurately predict the zero elements in dielectric tensors caused by the space group constraints.
> > > >
> > > > ------------------------

---

> > > > > ### Author Response · Authors · 2024-11-25
> > > > >
> > > > > **(3)** For practical application. The example in (2) is a simple illustration; when the space group of the input crystal is known, the prior knowledge in Appendix A.3 can be used to ensure that the network output fully obeys the constraints of tensor properties. For instance, the mask from Table 11 can be applied to weight the network output, ensuring 100\% compliance with the constraints in Table 11 (this can be applied during both the training and inference phases). This was not experimentally demonstrated in our current work, but we plan to explore related studies in future work.
> > > > >
> > > > > ----
> > > > >
> > > > > **Response to second question (round2)**
> > > > >
> > > > > On the dielectric dataset, our method shows an improvement of EwT 5\%, while for other metrics, our method indeed achieves results similar to the minimum frame averaging. Additionally, we have added the results for the piezoelectric tensor dataset and the elastic tensor dataset, as shown below:
> > > > >
> > > > > | Metric               | eCom. (Minimal Frame Averaging) | **GoeCTP (Ours)** |
> > > > > |----------------------|----------------------------------|-------------------|
> > > > > | **Fnorm ↓**          | 0.448                          | 0.448            |
> > > > > | **EwT 25% ↑**        | 44.5%                          | **44.9%**        |
> > > > > | **EwT 10% ↑**        | 42.7%                          | **43.1%**        |
> > > > > | **EwT 5% ↑**         | 37.3%                          | **40.1%**        |
> > > > > | **Total Time (s) ↓** | 999                            | 938              |
> > > > >
> > > > > *Table: Predictive performance comparisons between eComFormer (Minimal Frame Averaging) and GoeCTP on the piezoelectric dataset.*
> > > > >
> > > > > | Metric               | eCom. (Minimal Frame Averaging) | **GoeCTP (Ours)** |
> > > > > |----------------------|----------------------------------|-------------------|
> > > > > | **Fnorm ↓**          | 110.98                         | **107.11**       |
> > > > > | **EwT 25% ↑**        | **42.9%**                      | 42.5%            |
> > > > > | **EwT 10% ↑**        | 15.0%                          | **15.3%**        |
> > > > > | **EwT 5% ↑**         | 6.5%                           | **7.2%**         |
> > > > > | **Total Time (s) ↓** | 2488                           | 2422             |
> > > > >
> > > > > *Table: Predictive performance comparisons between eComFormer (Minimal Frame Averaging) and GoeCTP on the elastic dataset.*
> > > > >
> > > > > Our method performs better than minimum frame averaging on other datasets.
> > > > >
> > > > > ----------
> > > > >
> > > > > **$\bullet$  Response to Table 4 (round2)**
> > > > >
> > > > > For Table 4, the specific experimental steps are as follows:
> > > > >
> > > > > **(1)** Training GeoCTP, and after completion, we extract eComFormer from GeoCTP for comparative testing.
> > > > >
> > > > > **(2)** The original test dataset (i.e., the test dataset for the first and third columns) was adjusted to invariant positions using polar decomposition. The augmented test dataset (i.e., the test dataset for the second and fourth columns) was adjusted to random crystal positions using random orthogonal matrices.
> > > > >
> > > > > **(3)** Using the models obtained in Step 1, GeoCTP and eComFormer were tested on both the original test dataset and the augmented dataset to validate the effectiveness of GeoCTP.
> > > > > Since the original test dataset in Step 2 was adjusted to invariant positions, the processing is identical for GeoCTP. As a result, eComFormer and GeoCTP yield the same outcomes.
> > > > >
> > > > > If the original test dataset is not processed and eComFormer is trained individually, the results in Table 4 would look as follows:
> > > > >
> > > > > | Metric               | eCom. (ori. data) | eCom. (aug. data) | **GoeCTP (Ours)** (ori. data) | **GoeCTP (Ours)** (aug. data) |
> > > > > |----------------------|-------------------|-------------------|------------------------------|------------------------------|
> > > > > | **Fnorm ↓**          | 3.60             | 4.96             | 3.23                         | 3.23                         |
> > > > > | **EwT 25% ↑**        | 80.6%            | 69.4%            | 83.2%                        | 83.2%                        |
> > > > > | **EwT 10% ↑**        | 56.2%            | 42.7%            | 56.8%                        | 56.8%                        |
> > > > > | **EwT 5% ↑**         | 32.6%            | 18.5%            | 35.5%                        | 35.5%                        |
> > > > >
> > > > > *Table: Ablation study for verifying the \(O(3)\) equivariance with dielectric dataset.*
> > > > >
> > > > > We hope our response satisfactorily addresses the reviewer's concerns.
> > > > >
> > > > > --------
> > > > >
> > > > > **$\bullet$  Response to weakness 3 and 4 (round2)**
> > > > >
> > > > > Thank you for the nimble observation.
> > > > > This is indeed true. The experimental environment may influence the algorithm outcome. GMTNet uses an A100 GPU, while we used an RTX 3090. Besides, the versions of libraries such as PyTorch and Numpy used in experiments may differ from those used in GMTNet's experiments, potentially causing some discrepancies. To keep experimental environment same, we reproduce the experimental results by using the source code released by the corresponding authors and using the same setting as reported in their paper. Then according to our experimental results, our method is better to GMTNet.

---

> ### Comment · Reviewer_3tQt · 2024-11-25
> **Thank you for your further responses**
>
> Dear authors,
>
> Thank you again for providing further clarifications. My concerns regarding Table 4 have been resolved. Based on our discussions, it appears that the contribution, or at least the discussion detailed in the current paper, is somewhat limited. Specifically, the improvements of the proposed method seem to be influenced by the randomness of the training process. For instance, the marginal improvements beyond minimal frame averaging and the slightly mismatched GMTNet performances suggest limited advancements. And for the space group constraints, it is better that the model itself satisfy these constraints such that we can trust.
>
> This paper, in its current form, does not seem to address new challenges in this direction, based on the current discussions provided in the paper. Furthermore, the novelty of the method appears to be somewhat restricted. While I appreciate the additional experimental results you have shared, these concerns lead me to maintain my current score. I hope these issues can be considered in revisions or a more refined version of this paper in the future.
>
> Thank you.

---

### Official Review · Reviewer_RVFL · 2024-11-01

**Soundness:** 3
**Presentation:** 3
**Contribution:** 2
**Rating:** 5
**Confidence:** 5

**Summary:**

The authors propose an O(3)-equivariant framework, GoeCTP, for crystal tensor prediction. GoeCTP utilizes polar decomposition to rotate and reflect the crystal into a standardized invariant position in space. The orthogonal matrix obtained from the polar decomposition is used to achieve equivariant tensor property predictions. The GoeCTP method achieves higher quality prediction results and runs more than 13× faster on the elastic benchmarking dataset.

**Strengths:**

1. GoeCTP is plug-and-play as it can be readily integrated with any existing single-value property prediction network for predicting tensor properties.
2. GoeCTP does not introduce excessive computational overhead.

**Weaknesses:**

1. The article has limited contributions in terms of methodological innovation, as the methods and main structure used by the authors are derived from DiffCSP++[1] and Comformer[2]. For detail, the polar decomposition method used may have been inspired by DiffCSP++, while the code implementation adopts the structure of Comformer.
2. The article does not clearly explain why Equation 2 needs to be satisfied. It is suggested that the authors provide more background or explanation regarding the physical or mathematical significance of Equation 2 in relation to tensor property prediction. This would help readers better understand the importance of this equation within the proposed framework.
3. There are some citation issues in lines 339-340 of the article.


References:
[1] Rui Jiao, Wenbing Huang, Yu Liu, Deli Zhao, and Yang Liu. Space group constrained crystal generation. In The Twelfth International Conference on Learning Representations, 2024.
[2] Keqiang Yan, Cong Fu, Xiaofeng Qian, Xiaoning Qian, and Shuiwang Ji. Complete and efficient graph transformers for crystal material property prediction. In The Twelfth International Conference on Learning Representations, 2024.

**Questions:**

1. Is it the case that all tensor properties need to satisfy Equation 2, or only certain tensor properties? Why？Please provide specific examples of tensor properties, indicating which properties need to satisfy Equation 2 and which do not, along with an explanation of the underlying reasons for this distinction. This would help deepen the understanding of the method's applicability and limitations.

---

> ### Author Response · Authors · 2024-11-22
>
> We sincerely thank you for your valuable time and comments.
> Below we will address your questions in detail.
>
> ------------------
>
> **$\bullet$ Response to weakness 1**
>
> We somewhat get some inspiration from DiffCSP++. However, our method’s architecture is not derived from DiffCSP++. First of all, Polar decomposition is a classical mathematical method that was fully studied in the last century [1]; it was not proposed by DiffCSP++. The contribution of DiffCSP++ lies in applying polar decomposition to crystal generation tasks, where it uses polar decomposition and symmetric bases to obtain simplified and compressed original data.
> In contrast, we demonstrated that polar decomposition could be applied to achieve equivariant tensor property prediction, which has not been explored before.
> Additionally, since our method is plug-and-play and can be integrated with other prediction methods, our code implementation is based on other existing method.
> We hope our response addresses your concerns.
>
> [1] Higham N J. Computing the polar decomposition—with applications. *SIAM Journal on Scientific and Statistical Computing*, 1986, 7(4): 1160-1174.
>
>
> ------------------------
>
> **$\bullet$  Response to weakness 2 and question 1**
>
> Thank you for the reminder and suggestions. To enhance understanding, we have used the dielectric tensor as an example in Equation 2 in the paper. For different tensors, the form of Equation 2 may differ, and our method can be applied to different high-order tensors. We include additional explanations and clarifications in Appendix A.2.
>
>
>
> --------------------
>
> **$\bullet$  Response to weakness 3**
>
>
> Thank you for pointing this out. There are two methods named Crystalformer [1,2], but we made an incorrect citation. We correct this in the PDF.
>
> [1] Wang, Yingheng and Kong, Shufeng and Gregoire, John M and Gomes, Carla P. Conformal Crystal Graph Transformer with Robust Encoding of Periodic Invariance. *AAAI2024*.
>
> [2] Taniai, Tatsunori and Igarashi, Ryo and Suzuki, Yuta and Chiba, Naoya and Saito, Kotaro and Ushiku, Yoshitaka and Ono, Kanta. Crystalformer: Infinitely Connected Attention for Periodic Structure Encoding. *ICLR2024*.

---

> > ### Comment · Reviewer_RVFL · 2024-11-26
> >
> > Thank you to the authors for their response.
> >
> > Regarding weakness 1, the authors clarified that this work and DiffCSP++[1] build on existing methods but are applied to different tasks, which I find acceptable. They also emphasized that the approach is plug-and-play; however, the paper only provides results for its application to the Conformer model, without experimental evidence supporting its use with other models. In the appendix, the authors addressed weakness 2 and question 1, citing GMTNet[2] but without directly clarifying the applicability of Equation 2, leaving some contributions of the work unclear.
> >
> > Therefore, I maintain my previous score.
> >
> > References:
> > [1] Rui Jiao, Wenbing Huang, Yu Liu, Deli Zhao, and Yang Liu. Space group constrained crystal generation. In The Twelfth International Conference on Learning Representations, 2024.
> > [2] Keqiang Yan, Alexandra Saxton, Xiaofeng Qian, Xiaoning Qian, and Shuiwang Ji. A space group symmetry informed network for o (3) equivariant crystal tensor prediction. In Forty-first International Conference on Machine Learning, 2024b.

---

> > > ### Author Response · Authors · 2024-11-27
> > > **Response to Reviewer RVFL  (round2)**
> > >
> > > We sincerely thank you for reviewing again. We will attempt to address your concerns further.
> > >
> > > ----------
> > >
> > > **$\bullet$  Response to the application for other models (round2)**
> > >
> > > In the earliest version, we already presented this, as shown in Tables 15 and 16 in Appendix A.4. We demonstrated the results of our method combined with CrystalFormer [1] and iComFormer [2] (another method proposed in ComFormer). The details are as follows:
> > >
> > > | Metric                | MEGNET  | ETGNN   | GMTNet  | **GoeCTP (eCom.)** | **GoeCTP (iCom.)** | **GoeCTP (Crys.)** |
> > > |-----------------------|---------|---------|---------|--------------------|--------------------|--------------------|
> > > | **Fnorm ↓**          | 3.71    | 3.40    | 3.28    | **3.23**           | 3.40               | 3.53               |
> > > | **EwT 25% ↑**        | 75.8%   | 82.6%   | **83.3%** | 83.2%              | 81.7%              | 80.1%              |
> > > | **EwT 10% ↑**        | 38.9%   | 49.1%   | 56.0%   | **56.8%**          | 53.8%              | 52.9%              |
> > > | **EwT 5% ↑**         | 18.0%   | 25.3%   | 30.5%   | **35.5%**          | 32.3%              | 30.6%              |
> > > | **Total Time (s) ↓** | 663     | 1325    | 1611    | 616                | **535**            | 645                |
> > > | **Time/batch (s) ↓** | 0.052   | 0.104   | 0.126   | 0.048              | **0.042**          | 0.202              |
> > >
> > > *Table: Additional comparison of performance metrics on dielectric dataset.*
> > >
> > > | Metric                | MEGNET   | ETGNN   | GMTNet   | **GoeCTP (eCom.)** | **GoeCTP (iCom.)** | **GoeCTP (Crys.)** |
> > > |-----------------------|----------|---------|----------|--------------------|--------------------|--------------------|
> > > | **Fnorm ↓**          | 143.86   | 123.64  | 117.62   | 107.11             | **102.80**         | 107.44             |
> > > | **EwT 25% ↑**        | 23.6%    | 32.0%   | 36.0%    | 42.5%              | **46.7%**          | 43.5%              |
> > > | **EwT 10% ↑**        | 3.0%     | 3.8%    | 7.6%     | 15.3%              | **18.6%**          | 15.8%              |
> > > | **EwT 5% ↑**         | 0.5%     | 0.5%    | 2.0%     | 7.2%               | **8.2%**           | 7.9%               |
> > > | **Total Time (s) ↓** | 2899     | 4448    | >36000   | **2422**           | 4035               | 7891               |
> > > | **Time/batch (s) ↓** | 0.226    | 0.348   | >2.813   | **0.189**          | 0.315              | 0.616              |
> > >
> > > *Table: Additional comparison of performance metrics on elastic dataset.*
> > >
> > > [1] Taniai, Tatsunori and Igarashi, Ryo and Suzuki, Yuta and Chiba, Naoya and Saito, Kotaro and Ushiku, Yoshitaka and Ono, Kanta. Crystalformer: Infinitely Connected Attention for Periodic Structure Encoding. *ICLR2024*.
> > >
> > > [2] Yan, Keqiang and Fu, Cong and Qian, Xiaofeng and Qian, Xiaoning and Ji, Shuiwang. Complete and efficient graph transformers for crystal material property prediction. *ICLR2024*.
> > >
> > > ---------
> > >
> > > **$\bullet$  Response to Equation 2 (round2)**
> > >
> > > After citing GMTNet in the revised Appendix A.2, we explain the reasons. The details are as follows:
> > >
> > >
> > > **Dielectric tensor** The specific reason  is rooted in the fact that the dielectric tensor characterizes a material's polarization response to an external electric field, describing the relationship between the electric displacement $\mathbf{D}\in\mathbb{R}^3$ and the applied electric field $\mathbf{E}\in\mathbb{R}^3$ by $\mathbf{D}=\boldsymbol{\varepsilon}\mathbf{E}$.
> > > When an $O(3)$ group transformation $\mathbf{Q}$ is applied to the crystal structure, we have $\mathbf{D'}=\boldsymbol{\varepsilon}'\mathbf{E'}$, where $\mathbf{D'}=\mathbf{Q}\mathbf{D}$ and $\mathbf{E'}=\mathbf{Q}\mathbf{E}$. This results in the transformation of the dielectric tensor under $O(3)$ group transformation as $\boldsymbol{\varepsilon}'=\mathbf{Q}\boldsymbol{\varepsilon}\mathbf{Q}^T$.
> > >
> > > In order to better help the reader to understand the tensor properties of the crystal, we give the introduction and transformation rules of piezoelectric tensor as well as elastic tensor, which we add in Appendix A.2. The details
> > > are as follows:

---

> > > > ### Author Response · Authors · 2024-11-27
> > > >
> > > > **piezoelectric tensor** The piezoelectric tensor $\mathbf{e} \in \mathbb{R}^{3\times3\times3}$,
> > > > describes the relationship between the applied strain $\boldsymbol{\epsilon} \in \mathbb{R}^{3 \times 3}$ to the electric displacement field $\mathbf{D} \in \mathbb{R}^3$ within the material. Mathematically, this relationship is expressed as $\mathbf{D}\_i=\sum\_{jk}\mathbf{e}\_{ijk}\boldsymbol{\epsilon}\_{jk}$, with $i, j, k \in \{ 1, 2, 3 \}$.
> > > > When an $O(3)$ group transformation $\mathbf{Q}$ is applied to the crystal, the strain tensor and
> > > > electric displacement field is transformed to
> > > > $
> > > > \boldsymbol{\epsilon}\_{jk}^{\prime}=\sum\_{mn}\mathbf{Q}\_{jm}\mathbf{Q}\_{kn}\boldsymbol{\epsilon}\_{mn}$
> > > > and $\mathbf{D}\_{i}^{\prime}=\sum\_{\ell}\mathbf{Q}\_{i\ell}\mathbf{D}\_{\ell}$. The relation is then reformulated as $\mathbf{D}\_i^\prime=\sum_{jk}\mathbf{e}\_{ijk}^\prime\boldsymbol{\epsilon}\_{jk}^\prime $.
> > > >
> > > > Since $\mathbf{Q}$ is orthogonal matrix
> > > > ($\mathbf{Q}^{-1}=\mathbf{Q}^{T}$), we have
> > > > $\boldsymbol{\epsilon}\_{jk}=\sum\_{mn}\mathbf{Q}\_{mj}\mathbf{Q}\_{nk}\boldsymbol{\epsilon}^{\prime}\_{mn}$. Consequently,  $\mathbf{D}\_{i}^{\prime}$ can be represented as
> > > > \begin{equation}
> > > > \begin{aligned}
> > > > \mathbf{D}\_{i}^{\prime}&=\sum\_{\ell}\mathbf{Q}\_{i\ell}\mathbf{D}\_{\ell}\\\\
> > > > &=\sum\_{\ell}\mathbf{Q}\_{i\ell}\sum\_{jk}\mathbf{e}\_{\ell jk}\boldsymbol{\epsilon}\_{jk}\\\\
> > > > &=\sum\_{\ell}\mathbf{Q}\_{i\ell}\sum\_{jk}\mathbf{e}\_{\ell jk}(\sum\_{mn}\mathbf{Q}\_{mj}\mathbf{Q}\_{nk}\boldsymbol{\epsilon}^{\prime}\_{mn})\\\\
> > > > &=\sum\_{\ell}\mathbf{Q}\_{i\ell}\sum\_{mn}\mathbf{e}\_{\ell mn}(\sum\_{jk}\mathbf{Q}\_{jm}\mathbf{Q}\_{kn}\boldsymbol{\epsilon}^{\prime}\_{jk}) \quad (exchange \, sign, m \leftrightarrow j, n \leftrightarrow k)\\\\
> > > > &=\sum\_{jk}
> > > > \sum\_{lmn}\mathbf{Q}\_{il}\mathbf{Q}\_{jm}\mathbf{Q}\_{kn}\mathbf{e}\_{lmn}
> > > > \boldsymbol{\epsilon}\_{jk}^\prime
> > > > \end{aligned}
> > > > \end{equation}
> > > >
> > > > Therefore, under the $O(3)$ group transformation $\mathbf{Q}$, the transformed piezoelectric tensor $\mathbf{e}\_{ijk}^{\prime}$ is given by:
> > > > \begin{equation}
> > > > \mathbf{e}\_{ijk}^{\prime}=\sum\_{lmn}\mathbf{Q}\_{il}\mathbf{Q}\_{jm}\mathbf{Q}\_{kn}\mathbf{e}\_{lmn}.
> > > > \end{equation}
> > > >
> > > >
> > > > **Elastic tensor.**  The elastic tensor $C \in \mathbb{R}^{3\times3\times3\times3}$ relates the applied strain $\boldsymbol{\epsilon} \in \mathbb{R}^{3 \times 3}$ to the stress tensor $\sigma \in \mathbb{R}^{3 \times 3}$ within the material,
> > > > expressed as: $\boldsymbol{\sigma}\_{ij} =\sum\_{k \ell}C\_{ijk \ell}\boldsymbol{\epsilon}\_{k \ell}$, with $i,j,k,\ell \in \{1, 2, 3,4\}$.
> > > > When an $O(3)$ group transformation $\mathbf{Q}$ is applied to the crystal, the strain tensor and
> > > > stress tensor is transformed to
> > > > $
> > > > \boldsymbol{\epsilon}\_{jk}^{\prime}=\sum\_{mn}\mathbf{Q}\_{jm}\mathbf{Q}\_{kn}\boldsymbol{\epsilon}\_{mn}$
> > > > and $
> > > > \boldsymbol{\sigma}\_{jk}^{\prime}=\sum\_{mn}\mathbf{Q}\_{jm}\mathbf{Q}\_{kn}\boldsymbol{\sigma}\_{mn}$.
> > > > The relation becomes $\boldsymbol{\sigma}\_{ij}^{\prime} =\sum\_{k \ell}C\_{ijk \ell}^{\prime}\boldsymbol{\epsilon}\_{k \ell}^{\prime}$.
> > > >
> > > > Because $\mathbf{Q}$ is orthogonal matrix ($\mathbf{Q}^{-1}=\mathbf{Q}^{T}$), we have
> > > > $\boldsymbol{\epsilon}\_{jk}=\sum\_{mn}\mathbf{Q}\_{mj}\mathbf{Q}\_{nk}\boldsymbol{\epsilon}^{\prime}\_{mn}$, then $\boldsymbol{\sigma}\_{ij}^{\prime}$ can be represented as
> > > >
> > > >
> > > > \begin{equation}
> > > > \begin{aligned}
> > > > \boldsymbol{\sigma}\_{ij}^{\prime} &=\sum\_{mn}\mathbf{Q}\_{im}\mathbf{Q}\_{jn}\boldsymbol{\sigma}\_{mn}\\\\
> > > > &=\sum\_{mn}\mathbf{Q}\_{im}\mathbf{Q}\_{jn}\sum\_{pq}C\_{mnpq}\boldsymbol{\epsilon}\_{pq}\\\\
> > > > &=\sum\_{mn}\mathbf{Q}\_{im}\mathbf{Q}\_{jn}\sum\_{pq}C\_{mnpq}\sum\_{k\ell}\mathbf{Q}\_{kp}\mathbf{Q}\_{\ell q}\boldsymbol{\epsilon}^{\prime}\_{k\ell}\\\\
> > > > &=\sum\_{k\ell}\sum\_{mnpq    }\mathbf{Q}\_{im}\mathbf{Q}\_{jn}\mathbf{Q}\_{kp}\mathbf{Q}\_{lq}C\_{mnpq}\boldsymbol{\epsilon}^{\prime}\_{k\ell}
> > > > \end{aligned}
> > > > \end{equation}
> > > >
> > > > Therefore, under the $O(3)$ group transformation $\mathbf{Q}$, $C\_{ijkl}^{\prime}$ is represented as:
> > > > \begin{equation}
> > > > %f\_{rp}(\cdot) \to
> > > > C\_{ijkl}^{\prime}=\sum\_{mnpq}\mathbf{Q}\_{im}\mathbf{Q}\_{jn}\mathbf{Q}\_{kp}\mathbf{Q}\_{lq}C\_{mnpq}.
> > > > \end{equation}
> > > >
> > > > We're grateful for your feedback on our work. We hope our reply can address your concerns. We are happy to provide any additional clarification and discussion.

---

### Official Review · Reviewer_Q6Uc · 2024-11-07

**Soundness:** 3
**Presentation:** 2
**Contribution:** 3
**Rating:** 6
**Confidence:** 3

**Summary:**

This paper presents GoeCTP, a novel O(3)-equivariant framework for predicting tensor properties of crystalline materials. The key innovation is using polar decomposition to handle tensor equivariance through an external rotation and reflection (R&R) module, rather than building it into the network architecture.

**Strengths:**

1. Novel use of polar decomposition for handling tensor equivariance
2. Strong theoretical foundation with clear mathematical proofs
3. Clear illustrations and explanations of complex concepts

**Weaknesses:**

1. Limited discussion of potential limitations or failure cases
2. Only two datasets are used.1.

**Questions:**

1. What are the limitations of using polar decomposition for this application? Are there edge cases where it might not work well?

2. How does the method perform on different types of crystal structures beyond those tested?

---

> ### Author Response · Authors · 2024-11-22
>
> We sincerely thank you for your recognition of our method in terms of novel use of polar decomposition,  theoretical foundation, and clear concepts illustration. Below we will
> address your questions in detail.
>
> -------------
>
> **$\bullet $ Response to weakness 1 and question 1**
>
>
> We thank you for this helpful comment. We include additional limitation discussions in Appendix A.5.
> Specifically, for some cases, such as 2D crystals with single-layer structures [1,2], the rank of the lattice matrix $\mathbf{L}$ might be less than 3. Therefore, directly applying polar decomposition may not yield a unique $3 \times 3$ orthogonal matrix $\mathbf{Q}$ [3]. In this case, our method would fail to achieve the desired $O(3)$-equivariance.
> We hope our response addresses the reviewer’s question.
>
>
> [1] Novoselov, Kostya S and Jiang, Da and Schedin, F and Booth, TJ and Khotkevich, VV and Morozov, SV and Geim, Andre K. Two-dimensional atomic crystals. *Proceedings of the National Academy of Sciences*, 2005, 102(30): 10451-10453.
>
> [2] Sherrell, Peter C and Fronzi, Marco and Shepelin, Nick A and Corletto, Alexander and Winkler, David A and Ford, Mike and Shapter, Joseph G and Ellis, Amanda V. A bright future for engineering piezoelectric 2D crystals. *Chemical Society Reviews*, 2022, 51(2): 650-671.
>
> [3] Higham N J. Computing the polar decomposition—with applications. *SIAM Journal on Scientific and Statistical Computing*, 1986, 7(4): 1160-1174.
>
>
> -------------
>
> **$\bullet$  Response to weakness 2**
>
> For Weakness 2, we have added additional experiments using a new dataset --- the piezoelectric tensor dataset, which is from ref. [1]. The experimental results are as follows:
>
> |                          | MEGNET  | ETGNN   | GMTNet  | **GoeCTP (Ours)** |
> |--------------------------|---------|---------|---------|------------------|
> | **Fnorm ↓**              | 0.465   | 0.535   | 0.462   | **0.448**        |
> | **EwT 25% ↑**            | 43.9%   | 37.5%   | **45.7%** | 44.9%            |
> | **EwT 10% ↑**            | 37.9%   | 22.8%   | 39.3%   | **43.1%**        |
> | **EwT 5% ↑**             | 27.1%   | 13.8%   | 35.7%   | **40.1%**        |
>
> *Table: Predictive performance comparisons among MEGNET, ETGNN, GMTNet, and GoeCTP on the piezoelectric dataset.*
>
>
> Our method achieves optimal results on Fnorm, EwT 5\%, and EwT 10\% metrics.
>
> [1] Yan, Keqiang and Saxton, Alexandra and Qian, Xiaofeng and Qian, Xiaoning and Ji, Shuiwang. A Space Group Symmetry Informed Network for O (3) Equivariant Crystal Tensor Prediction. *ICML2024*.
>
>
>
> -------------
>
> **$\bullet$  Response to question 2**
>
> We are a bit confused about this question; perhaps you’re asking about the generality of our method. Currently, we have added a new piezoelectric dataset and still verify the effectiveness of our approach. If we find new publicly available datasets, we will continue to validate our method. Additionally, as we discussed in the limitations section, our method is theoretically generalizable to 3D crystals but may fail for 2D crystals. If our understanding of the question is incorrect, please feel free to point it out. Thank you very much.

---

> ### Comment · Reviewer_Q6Uc · 2024-11-26
>
> I acknowledge the efforts made by the authors.

---

### Author Response · Authors · 2024-11-22

We sincerely thank all four reviewers for their thoughtful feedback and insightful comments.
We are particularly encouraged by the reviewers’ feedback. We have made a heavy revision to our paper according to the reviewer's constructive suggestions. Below, we summarize some key modifications in the updated PDF document:

---------------------------------

*Main text*

$\bullet$  (Section 5.1) Add an introduction for new piezoelectric tensor
dataset

$\bullet$ (Section 5.2) Add comparison of prediction performance
on the piezoelectric dataset


$\bullet$ (Section 5.2) Add ablation study for verifying the $O(3)$ equivariance with piezoelectric dataset

$\bullet$ (Section 5.2) Add efficiency comparison on the piezoelectric dataset

---------------------------------

*Appendix*

$\bullet$ (Appendix A.2)
Add an introduction for $O(3)$ equivariance for different crystal tensor properties

$\bullet$ (Appendix A.3)
Add an introduction for the number of independent components in the piezoelectric tensor

$\bullet$ (Appendix A.5)
Add detailed discussion for limitations and extensibility of our method

$\bullet$ (Appendix A.6)
Add discussion for how to further utilize the tensor properties symmetry

---------------------------------

If further consideration remains, please kindly let us know. We are very happy to make a further revision in light of your great suggestions. We will address comments by each of the reviewers individually.

---

### Meta-Review · Area_Chair_6MrP · 2024-12-22

**Metareview:**

This paper received very weak support from the reviewers, and some of them raised major concerns on this paper.

**Additional Comments On Reviewer Discussion:**

Some of the major issues remain unresolved after rebuttals.

---

### Decision · Program_Chairs · 2025-01-22

Reject